# Deep Learning for SAR Ship Detection: Past, Present and Future

Jianwei Li [1], Congan Xu [1,2,*], Hang Su [1], Long Gao [1] and Taoyang Wang [3]

1   Information Fusion Institute, Naval Aviation University, Yantai 264000, China; lgm_jw@163.com (J.L.); shpersonal_email@163.com (H.S.); gaolong14@nudt.edu.cn (L.G.)
2   Advanced Technology Research Institute, Beijing Institute of Technology, Jinan 250300, China
3   School of Remote Sensing Information Engineering, Wuhan University, Wuhan 430000, China; wangtaoyang@whu.edu.cn
*   Correspondence: 7520220053@bit.edu.cn

**Abstract:** After the revival of deep learning in computer vision in 2012, SAR ship detection comes into the deep learning era too. The deep learning-based computer vision algorithms can work in an end-to-end pipeline, without the need of designing features manually, and they have amazing performance. As a result, it is also used to detect ships in SAR images. The beginning of this direction is the paper we published in 2017BIGSARDATA, in which the first dataset SSDD was used and shared with peers. Since then, lots of researchers focus their attention on this field. In this paper, we analyze the past, present, and future of the deep learning-based ship detection algorithms in SAR images. In the past section, we analyze the difference between traditional CFAR (constant false alarm rate) based and deep learning-based detectors through theory and experiment. The traditional method is unsupervised while the deep learning is strongly supervised, and their performance varies several times. In the present part, we analyze the 177 published papers about SAR ship detection. We highlight the dataset, algorithm, performance, deep learning framework, country, timeline, etc. After that, we introduce the use of single-stage, two-stage, anchor-free, train from scratch, oriented bounding box, multi-scale, and real-time detectors in detail in the 177 papers. The advantages and disadvantages of speed and accuracy are also analyzed. In the future part, we list the problem and direction of this field. We can find that, in the past five years, the $AP_{50}$ has boosted from 78.8% in 2017 to 97.8 % in 2022 on SSDD. Additionally, we think that researchers should design algorithms according to the specific characteristics of SAR images. What we should do next is to bridge the gap between SAR ship detection and computer vision by merging the small datasets into a large one and formulating corresponding standards and benchmarks. We expect that this survey of 177 papers can make people better understand these algorithms and stimulate more research in this field.

**Keywords:** SAR ship detection; SAR dataset; single-stage detector; two-stage detector; anchor free; train from scratch; oriented bounding box; multi-scale detection; deep learning; computer vision

## 1. Introduction

Synthetic aperture radar (SAR) remote sensing has become one of the important methods for marine monitoring due to its all-day, all-weather advantage. Ship detection in SAR images has broad prospects in both military and civilian fields [1,2].

The traditional detection method includes three steps: sea-land segmentation, CFAR (constant false alarm rate) detection, and discrimination [3,4]. In the sea-land segmentation step, the land pixels are rejected to avoid interference with the CFAR step. The common method is based on GIS (geographic information system) or image features. The gray histogram is the classical feature used for segmentation. In the second step, CFAR is usually used for ship detection. The distribution function is assumed to fit the pixel distribution of the SAR image. K, Weibull, and Rayleigh distribution are usually used in this step. To keep the probability of a false alarm at a constant value, the CFAR algorithm compares

the testing pixel with an adaptive threshold that is generated by the local background surrounding the testing pixel. After the pre-screening by CFAR, a discriminator is needed to reject the background. Discriminator includes two procedures: feature designing and classifier designing. According to the feature difference between ship chips and non-ship chips, this step can reduce the number of false alarms. The traditional detection method dominated this field for a long time.

With the development of deep learning-based object detection algorithms in computer vision (CV) [5], SAR researchers also began to seek inspiration from computer vision. There are three reasons that can explain the revival of deep learning. They are the arising of computing power, big data, and corresponding algorithms. As SAR images are not easily accessed, the deep learning-based detection method cannot be used for SAR ship detection at the beginning.

This problem was solved in 2017, as the first dataset SSDD (SAR Ship Detection Dataset) was open to the public. SSDD provides the same data and evaluation criteria for researchers, and it solves the problem that the traditional algorithms lack data and are not comparable in this field. Since then, more and more researchers adopt a deep learning-based method in this area. The deep learning-based algorithms also show great results compared with the traditional CFAR-based method. The active and open characteristics of computer vision also further promote the development of this field. We think that the emergence of SSDD means that this field comes into the deep learning era.

As far as we know, there are 177 papers [6–182] that use deep learning-based algorithms to detect ships in SAR images. However, there are no papers that review them yet. In order to summarize the achievements of the 177 papers and show the way for the future, we specially wrote this paper, hoping to contribute to the development of this field.

The rest of this review is arranged as shown in Figure 1. Section 2 briefly analyzes some work related to our paper. Section 3 summarizes the past of the traditional detection algorithms in SAR images. It mainly includes CFAR, hand-crafted features, and limited shallow representation. Section 4 introduces the present in deep learning-based detectors. We review the 177 papers, divide them into 10 categories and analyze them, respectively. Section 5 shows the future direction of this field. Section 6 is the conclusion of the paper.

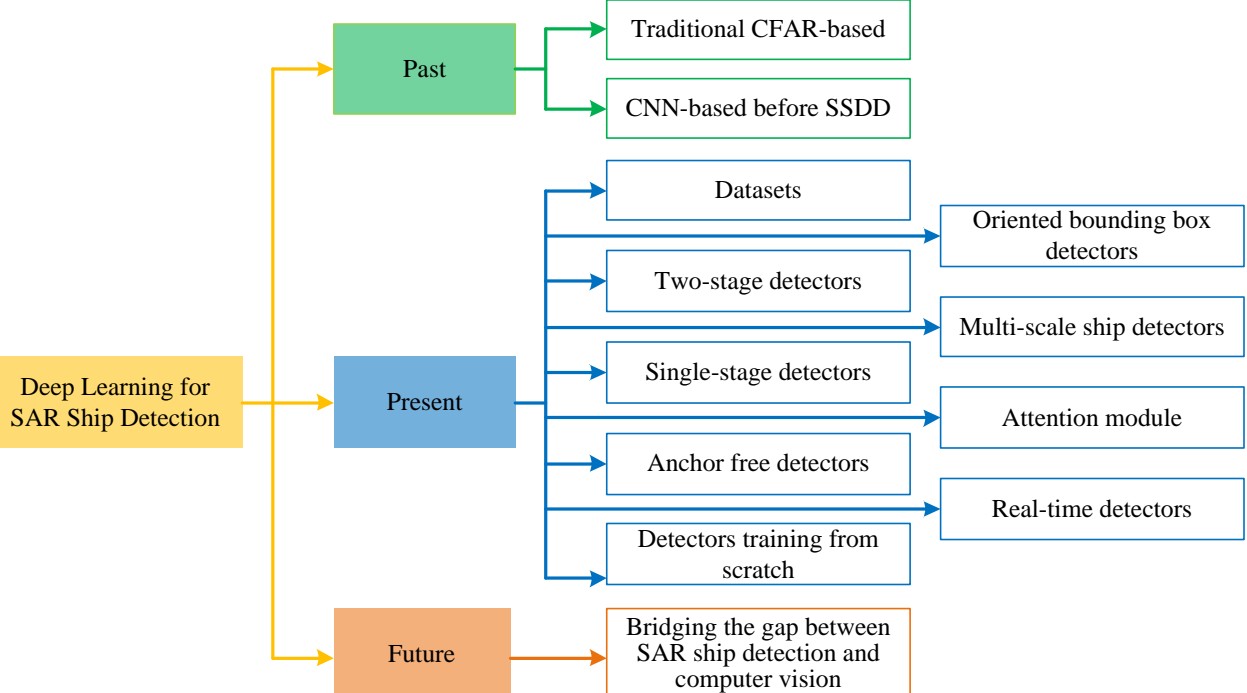

**Figure 1.** The overall architecture of the paper.

## 2. Related Work

As far as we know, few researchers have written review papers about this direction. This is partly due to the fact that this direction is new to some extent. At present, only three papers [105,127,170] have performed work related to our work.

Jerzy et al. [105] reviewed the papers from the last 5 years that discuss SAR ship detection. They mainly introduce the development of CFAR methods, CNN (convolutional neural network) based methods, GLRT (generalized likelihood ratio test) based methods, feature extraction-based methods, weighted information entropy-based methods, and variational Bayesian inference-based methods. Compared with paper [105], we mainly focus on the deep learning-based detection methods and do not focus on the traditional methods.

Mao et al. [127] solved the problem of the lack of performance benchmark for state-of-the-art methods on SSDD. Through this work, researchers can compare their work in the same experimental setup. They present 21 advanced detection models, including single-stage, two-stage, train from scratch detection algorithms, and so on. Compared with paper [127], we not only introduce the performance of different public datasets, but also classify all the papers, and summarize the principles and results of the algorithms.

Zhang et al. [170] solved the problem of the coarse annotations and ambiguous standards in SSDD. These improvements are beneficial for a fair comparison. It has played a great role in promoting the healthy development of this field. We suggest that researchers use the standards specified in this paper in the future. Compared with paper [170], our work is not limited to SSDD but introduces other datasets in this field. More importantly, our team has systematically analyzed, classified, and commented on the methods used, and pointed out the future research direction, which is beneficial to the development of this field.

In short, our work is different from the other papers. It is the first comprehensive review of SAR ship detection.

## 3. Past—The Traditional SAR Ship Detection Algorithms

Traditional detection algorithms in SAR images are based on hand-crafted features and limited shallow-learning representation. It can be divided into three steps: preprocessing, candidate region extraction, and discrimination.

CFAR is a common method for candidate region extraction. It can select potential ship regions. It first statistically models the clutter and then obtains the threshold value according to the false alarm rate. The pixels above the threshold are regarded as ship pixels, and those below the threshold are regarded as background. CFAR is essentially a segmentation-based algorithm, that is, the pixels are classified into two categories (ship or non-ship) according to the gray size, and then the ship pixel region is merged into the ship region. The performance of this method largely depends on the statistical modeling of sea clutter and the parameter estimation of the selected model. According to different SAR image products and practical application requirements, different statistical models such as Gaussian distribution, gamma distribution, log-normal distribution, Weibull distribution, and K distribution are proposed. Gaussian distribution and K distribution are the most commonly used. Generally speaking, when the scene is relatively simple, the CFAR method can achieve better results. However, for small ships and complex offshore scenes, due to the difficulty of modeling, it will have more false positives and poor detection performance.

Discrimination is generally realized by using artificially designed features and training classifiers. In addition to the simple features such as length, width, aspect ratio, and scattering point position, the features introduced from computer vision are also commonly used and have stronger robustness. Such as integral image features, HoG (histogram of oriented gradients), SURF (speeded up robust features), and LBP (local binary pattern). These features improve the performance of the detection algorithm. In classifier designing, decision trees, SVM, gradient boosting, and their improved versions also further improve the performance.

Feature and classifier designing have pushed this field forward in the past few years. However, since the rise of deep learning in 2012 [5], the above ideas are dwarfed in speed and accuracy. The object detection algorithm based on deep learning is an end-to-end processing method, as shown in Figure 2. It does not need to optimize multiple independent steps like the traditional method. It optimizes the whole detection system uniformly. It can adapt to various complex scenes (there is no need for sea–land segmentation in nearshore and port) and has very strong robustness. Therefore, in recent years, deep learning-based SAR ship detection algorithms have become a new research hotspot.

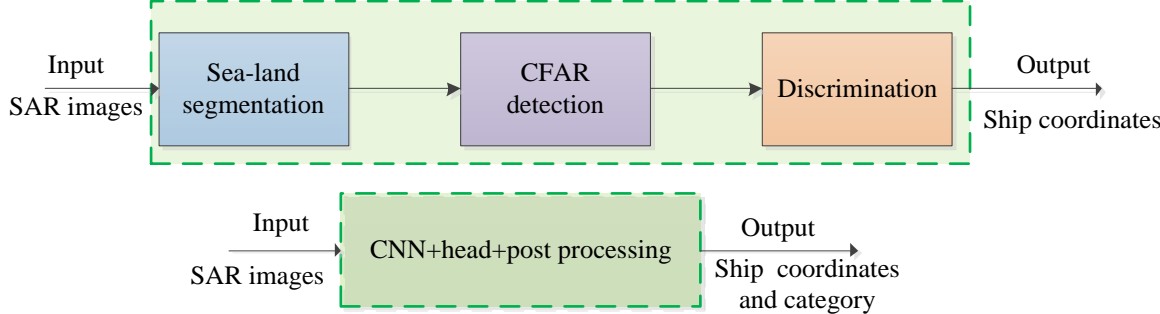

**Figure 2.** The differences between the CFAR-based detector and the deep learning-based detector.

The advantages and disadvantages of deep learning and traditional-based detection algorithms in SAR images can be proved by qualitative analysis or quantitative experiments.

The detection method based on CFAR has four main shortcomings through qualitative analysis. Firstly, CFAR needs to set the size of the protection window according to the size of the ship. It works well in the case of local uniform clutter in a single ship. If several ships with different sizes are close, the inconsistency between the change of ship size and the fixed protective window will lead to missed detection. Secondly, the CFAR algorithm needs to accurately model SAR images, which is difficult to implement. Thirdly, the essence of CFAR is an unsupervised algorithm, and its performance is essentially worse than the supervised algorithm (Faster R-CNN (region-based convolutional neural network), YOLO (you only look once), SSD (single shot detector), etc.). Fourthly, the CFAR algorithm and discrimination algorithm is a system pieced together after multiple links are debugged separately, and its performance cannot be compared with the end-to-end deep learning algorithm.

Sun Xian carried out a comparative experiment between the classical ship detection algorithm and the deep learning algorithm [65]. In the paper, the classical ship detection algorithms (optimal entropy automatic threshold method and CFAR method based on K distribution) are tested and analyzed on the AIR-SARShip-1.0 dataset. The experiment results are shown in Table 1. We can find that the performance of the deep learning algorithm is significantly better than that of the traditional algorithm.

**Table 1.** The performance difference between traditional-based detectors and deep learning-based detectors on the same condition [65].

| Algorithms | AP (Average Precision) |
|---|---|
| CFAR method based on K distribution | 19.2% |
| optimal entropy automatic threshold method | 28.2% |
| Faster R-CNN | 79.3% |
| SSD-512 | 74.3% |

Before SSDD, there are six papers that use convolutional neural networks [1–6] to detect ships in SAR images. We think that these six papers are not based on deep learning. The reasons are as follows. Firstly, some algorithms are not end-to-end, they just use CNN as a component in the traditional detection process. Secondly, although some algorithms

are end-to-end, the dataset and evaluation criteria are not public, which is difficult for future researchers to reproduce, and the results are also not comparable.

Due to the important role of SSDD, we take the publication date of SSDD paper as the time separation point between the traditional and deep learning-based detection algorithms. Therefore, we believe that ship detection in SAR images entered the era of deep learning on 1 December 2017, as shown in Figure 3. A large number of researchers gradually began to abandon the traditional detection algorithm based on CFAR and adopt the advanced detection method based on deep learning [6–182]. The overview of these deep learning-based detectors in SAR images is the focus of this paper.

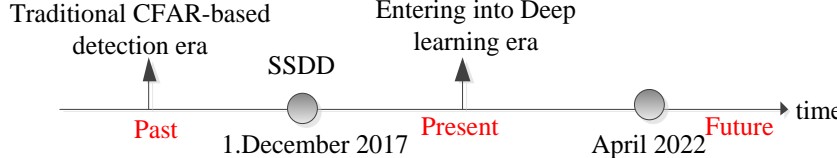

**Figure 3.** The time divisions of past, present and future.

## 4. Present—The Deep Learning-Based SAR Detection Algorithms

### 4.1. The General Overview of the 177 Papers

#### 4.1.1. The Countries

In the country view, we can find that 90% of the papers' authors are Chinese, which is shown in Figure 4. There is no doubt that Chinese researchers have been the mainstream in this direction. Several public datasets are constructed by Chinese researchers, which further prove the above opinion.

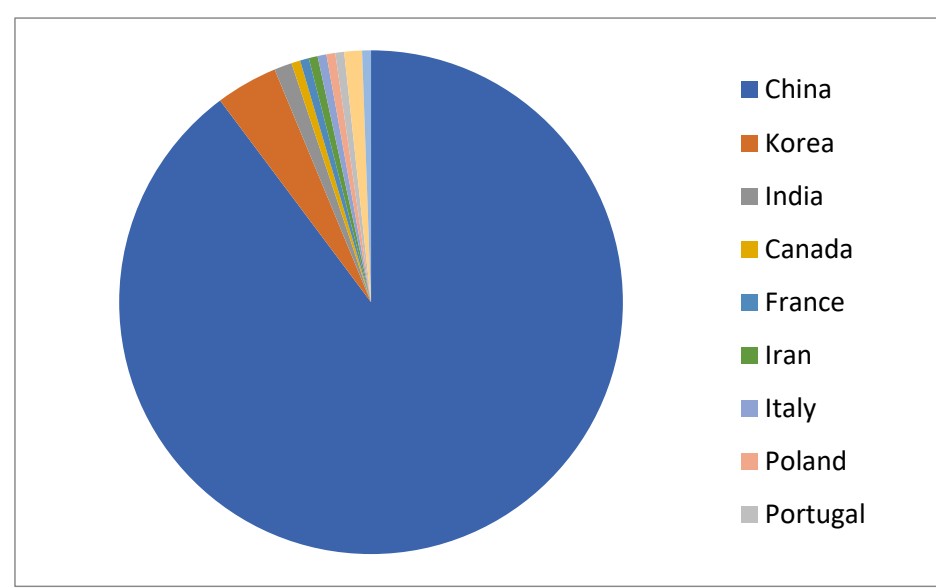

**Figure 4.** The percentages of the author countries.

#### 4.1.2. Journal or Conference

A total of 63% of the 177 papers are published in journals, and 37% are in conferences. The most common journals and conferences are Remote Sensing and IEEE International Geoscience and Remote Sensing Symposium (IGARSS), respectively.

#### 4.1.3. Timeline of the 177 Papers

The timeline of the deep learning based SAR ship detectors is shown in Figure 5.

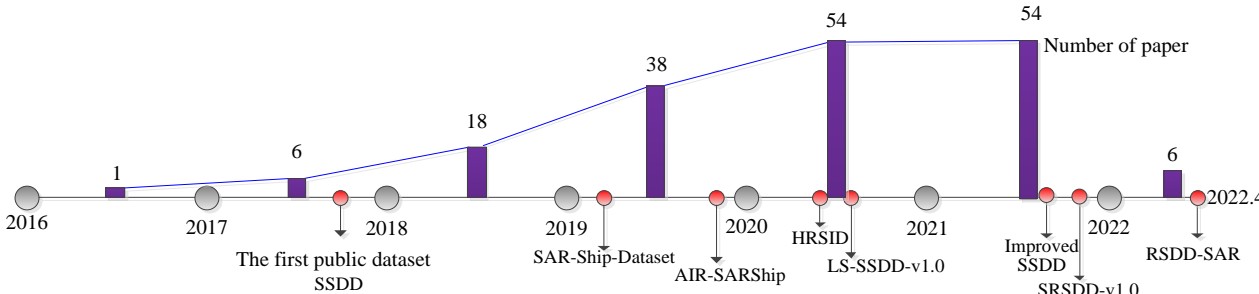

**Figure 5.** The timeline of the 177 papers.

Gray lines and gray circles in the figure represent the time, the purple bar represents the number of papers in the current year, and the red circles represent the public time of the dataset. From the timeline, we can find that in the passing five years, the number of papers about deep learning-based SAR ship detectors becoming more and more. The period 2016–2017 is in the transitional period between traditional and deep learning methods, and there are sporadic papers. Additionally, due to the lack of a unified dataset and the lack of in-depth understanding of deep learning and computer vision algorithms, the application of deep learning algorithms is not thorough. This situation did not change until the emergence of the first dataset (SSDD) paper at the end of 2017. SSDD discloses the data and evaluation criteria it used, which lays the foundation for the rapid development of this field. Since then, the fast lane of research was opened, and a large number of papers were published. The milestones of this field are the several open-access datasets, which are shown in the red circle above. With the increase in available datasets, more and more researchers are paying attention to this field.

### 4.1.4. The Datasets and Satellites That Are Used

The datasets that those papers used are shown in Table 2. We can see that SSDD is the most frequently used dataset for now. It was used 83 times, 62.4% percent of the total. Additionally, the usage of several public datasets shows a gradual upward trend.

**Table 2.** The datasets that are used in the past five years.

| Datasets | 2016 | 2017 | 2018 | 2019 | 2020 | 2021 | 2022 | Total |
|---|---|---|---|---|---|---|---|---|
| SSDD | 0 | 1 | 2 | 19 | 28 | 29 | 4 | 83 |
| SSDD+ | 0 | 0 | 1 | 2 | 1 | 2 | 0 | 6 |
| SAR-Ship-Dataset | 0 | 0 | 0 | 1 | 4 | 14 | 1 | 20 |
| AIR-SARShip1.0/2.0 | 0 | 0 | 0 | 1 | 3 | 5 | 0 | 9 |
| HRSID | 0 | 0 | 0 | 0 | 2 | 6 | 1 | 9 |
| LS-SSDD-v1.0 | 0 | 0 | 0 | 0 | 1 | 1 | 1 | 3 |
| Official-SSDD | 0 | 0 | 0 | 0 | 0 | 1 | 0 | 1 |
| SRSD-v1.0 | 0 | 0 | 0 | 0 | 0 | 1 | 0 | 1 |
| RSDD-SAR | 0 | 0 | 0 | 0 | 0 | 0 | 1 | 1 |
| Total | 0 | 1 | 3 | 23 | 40 | 58 | 8 | 133 |

Before the first dataset paper was published in 2017, different researchers adopted different SAR images and indicators to test their detectors. Thus, the results of the papers are not comparable. This phenomenon is not beneficial for the development of this field. In order to overcome this, we constructed the first dataset SSDD and it is open to the public. Meanwhile, we provide another dataset called SSDD+, which shares the same images with SSDD but has an oriented bounding box. With the rapid development of deep learning-based computer vision algorithms after 2019, SSDD draws more attention to researchers. Zhang analyzed the usage situation of SSDD in paper [170]. From this paper, we can find that SSDD becomes the most popular dataset, though it has many drawbacks. In addition to this, SAR-Ship-Dataset and AIR-SARShip are showing great potential to be

the popular dataset. The other datasets were seldom used, as their public dates are a little late. As deep learning models need more data to prevent overfitting, the future of this field is to merge them into a large dataset and provide the benchmarks on the large dataset with the common detection algorithms in computer vision. We think that, if the dataset is big enough, the benchmark is whole enough, and the maintenance is regular enough, it will be accepted by most researchers. This is the focus of our future work.

Table 3 shows the SAR satellites that papers used besides the public dataset. We can find that SAR images from Sentinel-1 are the most frequently used all the time. This is because the data are easy to acquire, and can be downloaded for free.

**Table 3.** The satellites that are used in the paper beside the datasets.

| Satellites | 2016 | 2017 | 2018 | 2019 | 2020 | 2021 | 2022 | Total |
|---|---|---|---|---|---|---|---|---|
| Sentinel-1 | 1 | 4 | 7 | 6 | 6 | 2 | 0 | 26 |
| RadarSat-2 | 1 | 0 | 2 | 2 | 2 | 0 | 0 | 7 |
| ALOS PALSAR | 0 | 1 | 0 | 0 | 0 | 0 | 0 | 1 |
| TerraSAR-X | 0 | 1 | 0 | 0 | 3 | 0 | 0 | 4 |
| Gaofen-3 | 0 | 1 | 5 | 6 | 5 | 2 | 1 | 20 |
| COSMO_SKYMed | 0 | 0 | 1 | 0 | 2 | 0 | 0 | 3 |
| AISSAR | 0 | 0 | 1 | 0 | 0 | 0 | 0 | 1 |

However, as China's first C-band multi-polarization SAR satellite Gaofen-3 was officially put into use on 23 January 2017, the policy of obtaining Gaofen-3images has become easier and easier. More and more papers use Gaofen-3 as the source image.

### 4.1.5. Deep Learning Framework

A deep learning framework can reduce the workload of researchers [183–187]. So, since the emergence of CAFFE (Convolution Architecture For Feature Extraction) [188] in 2017, it gets more and more attention from researchers. Table 4 shows the deep learning framework those 177 papers used. We can find that in the beginning years (2017–2018), CAFFE is the most frequently used framework. It is because CAFFE is the first common deep learning framework that researchers use and most of the detection algorithms in computer vision are based on CAFFE, for example, Faster R-CNN and SSD. In order to improve the efficiency of the deep learning framework, Google provided TensorFlow [189] in 2017. Compared with CAFFE, Tensorflow is more powerful and easier to use. A lot of researchers adopt Tensorflow as their framework. After Tensorflow, PyTorch [190] was promoted by Facebook FAIR in 2017. PyTorch is more suitable for researchers, and the number of users surpasses the Tensorflow gradually. In addition to CAFFE, Tensorflow and Pytorch, Keras, DarkNet and PaddlePaddle are also used by some researchers. Due to the fact that most detection algorithms in computer vision are based on Tensorflow and Pytorch, we recommend researchers in this area use them as the deep learning framework.

**Table 4.** The deep learning framework those papers used.

| Framework | 2016 | 2017 | 2018 | 2019 | 2020 | 2021 | 2022 | Total |
|---|---|---|---|---|---|---|---|---|
| Caffe | 0 | 3 | 9 | 3 | 6 | 2 | 0 | 23 |
| Tensorflow | 0 | 2 | 3 | 12 | 5 | 7 | 0 | **29** |
| Pytorch | 0 | 0 | 0 | 3 | 19 | 18 | 6 | **44** |
| Keras | 0 | 0 | 0 | 1 | 3 | 3 | 0 | 7 |
| DarketNet | 0 | 0 | 0 | 0 | 1 | 3 | 0 | 4 |
| PaddlePaddle | 0 | 0 | 0 | 0 | 0 | 1 | 0 | 1 |

### 4.1.6. Performance Evolution

Tables 5–11 show the performance of several public datasets. In the 'AP' column, the large number represents $AP_{50}$, and the small one represents AP. AP50 refers to the average

precision with IoU (intersection over union) = 50%. AP refers to the value of IoU from 50% to 95% in steps of 5% and then calculates the average value of AP under these IoUs. Normally, $AP_{50}$ is higher than AP. $AP_{50}$ and AP are usually used in PASCAL VOC and MS COCO, respectively. Since the dataset contains only one class of ships, the mAP value is the same as the AP value.

In the table, the italics represent the performance of two-stage detectors, and the non-italics represent the performance of single-stage detectors. The blue, red, green, purple, and golden colors represent anchor-free, train from scratch, oriented bounding box, multi-scale, and attention detectors. The underlines represent real-time detectors.

The number of papers in Tables 5–11 is less than in Table 2. This is because some of the papers in Table 2 did not use the AP or $AP_{50}$ as the evaluation indicator, so we do not show them in Tables 5–11.

From Table 5, we can see that there are 52 papers that are trained and tested on SSDD. Additionally, in the past five years, $AP_{50}$ of detectors on SSDD boosted from 78.8% in 2017 to 97.8 % in 2022. The testing time is also getting faster and faster. What should be noticed is that as the train-test division is ambiguous in the original SSDD, so the AP in Table 5 is not comparable to some extent. That is also why we recommend the following researchers adopt the Improve SSDD [170] as the new standard.

**Table 5.** The performance evolution of detectors on SSDD (The data come from the 177 papers).

| No. | Date | AP | Time | No. | Date | AP | Time |
|---|---|---|---|---|---|---|---|
| *11* | *1 December 2017* | *78.8%* | *173 ms* | **104** | **14 October 2020** | **92.6%** **56.5%** | **7.39 ms** |
| **15** | **9 March 2019** | 91.3% | **96 ms** | 111 | 16 November 2020 | 91.84% | |
| 39 | 2 April 2019 | 89.76% | 10.938 ms | *115* | *2 December 2020* | *89.79%* | |
| **40** | **21 May 2019** | **90.16%** | **21 ms** | 116 | 3 December 2020 | 90.7% | 13.6 ms 74 FPS |
| **43** | **24 July 2017** | **79.78%** | **28.4 ms** | 117 | 4 December 2020 | 88.33% | 15 FPS |
| **51** | **23 September 2019** | **80.12%** | **9.28 ms** | **118** | **7 December 2020** | **94.6%** | **3.9 ms** **258 FPS** |
| **54** | **24 October 2019** | **94.13%** | **9.03 ms** **111 FPS** | 121 | **28 December 2020** | **95.1%** | **33 ms** |
| 55 | 24 October 2019 | 90.04% | 87 ms | **131** | **12 February 2021** | **95.7%** **63.4%** | |
| 56 | 14 November 2019 | 94.7% | | **132** | **17 February 2021** | **93.78%** | **202 FPS** |
| *62* | *14 November 2019* | *83.4%* | | 134 | 27 February 2021 | 80.45% | |
| *63* | *14 November 2019* | *90.44%* | *96.04 ms* | 149 | 17 March 2021 | 94.41% | 31 FPS |
| 68 | December 2019 | 96.93% | 8.72 ms | 146 | 23 March 2021 | 95.52% | |
| *69* | *2 January 2020* | *97.9%* *64.6%* | *103 ms* | 148 | 31 March 2021 | 92.09% | |
| 74 | 19 March 2020 | 96.4% 67.4% | 106.4 ms | 151 | 13 May 2021 | 88.08% | 12.25 ms |
| *78* | *30 March 2020* | *94.2%* *59.5%* | *0.93 M* | 154 | 9 June 2021 | 98.4% | |
| **81** | **3 April 2020** | **94%** | | **157** | 30 June 2021 | 61.4% | 45 FPS |
| 82 | 16 April 2020 | 90.08% 68.1% | | **158** | **1 July 2021** | **96.8%** **62.7%** | **438 ms** |
| **84** | **22 April 2020** | **97.07%** | **233 FPS** | *160* | *13 July 2021* | *97.2%* *61.5%* | |
| 90 | 25 May 2020 | 93.96% | | **161** | **14 July 2021** | **95.29%** | **11 FPS** |
| 93 | 24 June 2020 | 94.72% | 63.2 ms | 170 | 6 December 2021 | 97.8% 64.9% | |
| **96** | **21 July 2020** | **96.08%** | **4.51 ms** **222 FPS** | 171 | 10 December 2021 | 82.2% | 5.2 ms |
| **98** | **21 August 2020** | **81.17%** | **24 ms** | 173 | 22 December 2021 | 97.4% | 42.5 FPS |
| 99 | 21 August 2020 | 83.4% | | 174 | 6 February 2022 | 95.6% 61.1% | |
| **100** | **31 August 2020** | **90.57%** | **17.2 ms** | 175 | 25 February 2022 | 97.8 % | 17.5 FPS |
| **102** | **6 October 2020** | **86.3%** | | 176 | 25 February 2022 | 95.03% | 47 FPS |
| **103** | **14 October 2020** | **95.6%** **61.5%** | | 177 | 19 March 2022 | 97.0% | |

From Table 6, we can see that there are only four papers that are trained and tested on SSDD+, and the $AP_{50}$ performance is increased from 84.2% in 2018 to 94.46% in 2021. The overall performance is a bit lower than that of SSDD. That is because the detectors on SSDD+ should predict an additional parameter (angle). We also find that the SSDD+ is seldom used compared with SSDD. That is, few researchers are interested in oriented bounding box detection in this area.

**Table 6.** The performance evolution of detectors on SSDD+ (the data come from the 177 papers).

| No. | Date | AP | Time |
|---|---|---|---|
| 20 | 29 August 2018 | 84.2% | 40 FPS |
| 41 | 26 June 2019 | 81.36% | |
| 83 | 20 April 2020 | 90.11% | 62.77 ms |
| 124 | 8 January 2021 | 94.46% | |

From Table 7, we can see that there are 14 papers that are trained and tested on SAR-Ship-Dataset, and the $AP_{50}$ performance is boosted from 89.07% in 2019 to 96.1% in 2021. The running speed is also accelerated to 60.4 FPS with 96.1% AP. The overall performance is a bit lower than that of SSDD. That is because this dataset is relatively larger than SSDD.

**Table 7.** The performance evolution of detectors on SAR-Ship-Dataset (the data come from the 177 papers).

| No. | Date | AP | Time | No. | Date | AP | Time |
|---|---|---|---|---|---|---|---|
| 38 | 29 March 2019 | 89.07% | | 138 | 17 February 2021 | 92.4% | |
| 89 | 20 May 2020 | 94.7% | 18 ms | 157 | 19 May 2021 | 93.46% | 339 FPS |
| 113 | 30 November 2020 | 91.89% | 12.05 FPS | 158 | 8 June 2021 | 95.52% | |
| 114 | 30 November 2020 | 91.07% | | 163 | 1 July 2021 | 95.8% | |
| 123 | 5 January 2021 | 90.25% | 22 ms | 166 | 14 July 2021 | 94.39% | |
| 133 | 17 February 2021 | 93.9% | | 178 | 22 December 2021 | 96.1 | 60.4 FPS |
| 136 | 17 February 2021 | 95.1% | | 179 | 6 February 2022 | 95.1 | |

From Table 8, we can see that there are only four papers are trained and tested on AIR-SARShip, and the $AP_{50}$ performance is boosted from 88.01% in 2019 to 92.49% in 2021. In addition, the running speed becomes 7.98 times faster (from 41.6 ms to 5.22 ms). The overall performance is a bit lower than that of SSDD.

**Table 8.** The performance evolution of detectors on AIR-SARShip (the data come from the 177 papers).

| No. | Date | AP | Time | Version |
|---|---|---|---|---|
| 65 | 1 December 2019 | 88.01% | 24 FPS | 1.0 |
| 97 | 13 August 2020 | 86.99% | | 1.0 |
| 130 | 8 February 2021 | 80.9% | | 1.0 |
| 171 | 1 December 2021 | 92.49% | 5.22 ms | 2.0 |

From Table 9, we can see that there are only nine papers are trained and tested on HRSID, and the $AP_{50}$ performance is boosted from 89.3% in 2019 to 94.4% in 2021. The overall performance is a bit lower than that on SSDD. That is because this dataset is relatively larger than SSDD.

**Table 9.** The performance evolution of detectors on HRSID (the data come from the 177 papers).

| No. | Date | AP | No. | Date | AP |
|-----|------|-----|-----|------|-----|
| 94 | 29 June 2020 | 89.3% 69.4% | 168 | 6 August 2021 | 89.2% 68%% |
| *110* | *10 November 2020* | *not given 84.4%* | 174 | 14 February 2022 | 91.4% 66.4% |
| 120 | 23 December 2020 | 91.99% 68.5% | 175 | 6 December 2021 | 94.4% 72% |
| 131 | 12 February 2021 | 92.4% 69.5% | 178 | 22 December 2021 | 88.3% |
| <u>**165**</u> | <u>**13 July 2021**</u> | <u>**90.7%**</u> <u>**69.4%**</u> | | | |

From Table 10, we can see that there are only three papers are trained and tested on LS-SSDD-v1.0, and the AP performance is boosted from 72.3% in 2019 to 75.5% in 2022. The overall performance is a bit lower than that on SSDD. LS-SSDD-v1.0 is specially used for large-scale SAR ship detection, which is fit for satellite-based SAR systems. It should be used more in the future.

**Table 10.** The performance evolution of detectors on LS-SSDD-v1.0 (the data come from the 177 papers).

| No. | Date | AP |
|-----|------|-----|
| 101 | 15 September 2020 | 75.3% |
| 168 | 6 August 2021 | 71.7% |
| 180 | 25 February 2022 | 75.5% |

The above datasets are relatively smaller than the datasets used in computer vision. In order to improve the generalization ability of the detector, researchers should use a large dataset. Some researchers merge several datasets into a large one as shown in Table 11. From Table 11, we can see that there are three papers that are trained and tested on the composite dataset, and the AP performance is 81.13%, 71.4%, and 95.1%, respectively. As deep learning-based detectors are data-hungry, we should merge the public datasets into a large one to prevent over-fitting.

**Table 11.** The performance evolution of detectors on other datasets (the data come from the 177 papers).

| No. | Date | AP$_{50}$ | Time | Datasets |
|-----|------|-----------|------|----------|
| **108** | **30 October 2020** | **81.13%** | **35.5 ms** | **SSDD + SAR-Ship-Dataset** |
| 125 | 27 January 2021 | 71.4% | 2920 ms | SAR-Ship-Dataset +AIRSAR-Ship-1.0 |
| **167** | **26 July 2021** | **95.1%** | | **HRSID + SSDD + IEEE 2020 Gaofen Challenge** |

### 4.2. The Algorithm Taxonomy of the 177 Papers

We divide the 177 papers into 10 categories, they are papers about datasets, two-stage detectors, single-stage detectors, anchor-free detectors, train from scratch detectors, detectors with the oriented bounding box, multi-scale detectors, detectors with attention module, real-time detectors, and others. The percentages of each algorithm are shown in Table 12. What should be explained is that the summation of the percentages is larger than 1. This is because many algorithms in the papers have several attributes. For example, it not only belongs to the single-stage detector but is also trained from scratch.

**Table 12.** The percentage of each algorithm.

| Algorithms | Datasets | Two-Stage | Single-Stage | Anchor Free | Scratch |
|---|---|---|---|---|---|
| Percentage | 5% | 26.7% | 25.6% | 5.1% | 4.0% |
| **Algorithms** | **Oriented** | **Multi-scale** | **Attention** | **Real-time** | **Others** |
| Percentage | 5.7% | 14.2% | 5.1% | 13.1% | 14.2% |

From Table 12, we can find the following conclusions. Firstly, there are eight papers that introduce the datasets to the researchers. They make a great contribution to this field. Secondly, two-stage detectors used in this field are slightly more than single-stage detectors. This is partly because the two-stage detectors have higher accuracy than the single-stage detectors in most cases. In addition, accuracy is the first consideration at the moment. Thirdly, anchor-free detectors, detectors trained from scratch, oriented bounding box detectors, and detectors with attention modules almost have a percentage of 5–6% among the 177 papers. This is because, as the above four directions are rare, they are not yet noticed by many researchers. In fact, these directions can overcome the problems of the ship size distribution abnormal and the lack of SAR images. They should be paid more attention in the future. Fourthly, almost 14% of papers are about multi-scale SAR ship detection, which is a little higher than other directions. This is because, compared with objects in computer vision images, ships in SAR are rather small. In order to improve the performance, detectors should pay more attention to multi-scale ships. Fifthly, 14.20% of papers are classified as others, which represents that these papers do not belong to the nine categories. Sixthly, only three papers, which is 1.7% of the 177 papers are reviewed in this field. Considering the active research in this field, it is not enough for now. This is one of the motivations for our work.

*4.3. The Public Datasets*

4.3.1. Overview

As far as we know, there are 10 public datasets that are used for training and detecting ships in SAR images. They are SSDD(SSDD+) [11], SAR-Ship-Dataset [38], AIR-SARShip1.0 [65], HRSID [94], LS-SSDD-v1.0 [101], AIR-SARShip2.0 [191], Official-SSDD [170], SRSDD-v1.0 [177] and RSDD-SAR [192]. Table 13 shows the detailed information of the 10 public datasets, in which the annotations of SSDD+, Official-SSDD, SRSDD-v1.0, and RSDD-SAR are the oriented bounding box.

**Table 13.** Detail information of existing public datasets.

| Dataset | Date | Source | Resolution | Image Size | Images/Ships | Annotation |
|---|---|---|---|---|---|---|
| SSDD (SSDD+) | 1 December 2017 | RadarSat-2 TerraSAR Sentinel-1 | 1 m–15 m | 190–668 | 1160/2456 | vertical oriented |
| SAR-Ship-Dataset | 29 March 2019 | Gaofen-3 Sentinel-1 | 3 m–25 m | 256 × 256 | 43,918/59,535 | vertical |
| AIR-SARShip-1.0 AIR-SARShip-2.0 | 1 December 2019 25 August 2021 | Gaofen-3 | 1 m, 3 m | 3000 × 3000 1000 × 1000 | 31 300 | vertical |
| HRSID | 29 June 2020 | Sentinel-1 TerraSAR | 0.5 m, 1 m, 3 m | 800 × 800 | 5604/16,951 | polygon |
| LS-SSDD-v1.0 Official-SSDD | 15 September 2020 15 September 2021 | Sentinel-1 | 5 m, 20 m | 24,000 × 16,000 The same as SSDD | 15/6015 | vertical polygon oriented |
| SRSDD-v1.0 | 15 December 2021 | Gaofen-3 | 1 m | 1024 × 1024 | 666/2275 | recognition |
| RSDD-SAR | April 2022 | Gaofen-3 TerraSAR | 2–20 m | 512 × 512 | 7000/10,263 | oriented |

In addition to these, SMCDD [182] is a good dataset based on China's first commercial SAR satellite HISEA-1. It has 1851 bridges, 39,858 ships, 12,319 oil tanks, and 6368 aircraft. It shows a great advantage in multi-class ship detection.

In the future, it is very necessary to combine the above datasets into a large one to avoid the problem of overfitting.

In the following part, we will introduce the details of the datasets and evaluate their advantages and drawbacks.

### 4.3.2. SSDD, SSDD+ and Official-SSDD

We made our dataset SSDD publicly available at the conference of 2017BIGSARDATA in Beijing [11]. SSDD is the first open dataset in this community. It can be a benchmark for researchers to train and evaluate their algorithms. In SSDD, there are a total of 1160 images and 2456 ships. The ships in SSDD have rich diversity, including small-size ships, complex backgrounds, and dense arrangements near the wharf. We also give the statistical results of the length, width, and aspect ratio of the ship bounding box in SSDD. The papers that used SSDD and their performance are shown in Table 5.

At the same time, based on 1160 SAR images of SSDD, we use the oriented bounding box to relabel the ship and obtain the dataset SSDD+. SSDD+ is the first dataset for SAR ship detection with an oriented bounding box. The papers that used SSDD+ and their performance are shown in Table 6.

At that time, there were some problems in SSDD due to the lack of understanding of computer vision and deep learning. The drawbacks of SSDD are the coarse annotations and ambiguous standards of use. It hinders fair comparisons and effective academic exchanges in this field.

In September 2021, Zhang [170] systematically analyzed and improved the problem of SSDD; they call it Official-SSDD. Zhang relabeled ships in SSDD and proposed three new datasets; they are bounding box SSDD, rotatable bounding box SSDD, and polygon segmentation SSDD. In addition, they also formulate some standards: the train-test division, the inshore-offshore protocol, the ship-size definition, the determination of the densely distributed small ship ships, and the determination of the densely parallel berthing at ports ship samples. We suggest that follow-up researchers use the Official-SSDD and standards proposed in paper [170] to carry out their relevant research.

### 4.3.3. SAR-Ship-Dataset

The training of the deep learning model depends on a large amount of data, and the amount of SSDD is relatively small. To solve this problem, Wang Chao [38] constructed a dataset called SAR-Ship-Dataset. SAR-Ship-Dataset contains 43,819 images and 59,535 ships, which are more than SSDD. The sources of SAR-Ship-Dataset are 102 Gaofen-3 images and 108 Sentinel-1 SAR images. These ships have distinct scales and backgrounds. The resolution, incident angle, polarization mode, and imaging mode are also diverse, which are helpful for the deep learning models to fit different conditions. The papers that used SAR-Ship-Dataset and their performance are shown in Table 7.

### 4.3.4. AIR-SARShip

A dataset containing more diverse scenes and covering various types of ships will help to train a model with better performance, stronger robustness, and higher practicability. In order to achieve the above purpose, Sun Xian constructed a dataset based on the Gaofen-3 satellite, named AIR-SARShip-1.0 [65].

It contains a total of 31 large images. A total of 21 images are training data and the other 10 images are testing data. The image resolutions include 1 m and 3 m. The image size is about 3000 × 3000 pixels. The information of each image in the dataset includes image number, pixel size, resolution, sea state, scene, and the number of ships. The dataset has the characteristics of a large scene and a small ship.

On the basis of version 1.0, Sun Xian and other researchers added more Gaofen-3 data to build AIR-SARShip-2.0. The dataset contains 300 SAR images. The scene types include ports, islands, reefs, sea surfaces with different levels of sea conditions, etc. The annotation information includes the location of ships, which has been confirmed by professional interpreters.

The papers that used AIR-SARShip and their performance are shown in Table 8.

### 4.3.5. HRSID

The original SAR image used to construct HRSID [94] includes 99 Sentinel-1B images, 36 TerraSAR-X images, and 1 TanDEM-X image. HRSID has 5604 high-resolution SAR images with 800 × 800 pixels to meet the needs of actual training for GPU. It is designed for ship detection and segmentation based on CNN, and it only contains one category of ships. It is divided into 65% training set and 35% testing set. It uses polygons to label the ship. In order to reduce the deviation of the ship detection algorithm, the interference derived from the ship is marked as a part of the ship.

According to statistics, the total number of ships marked in HRSID is 16,951, and each SAR image contains an average of three ships. The number of small ships, medium ships, and large ships accounted for 54.5%, 43.5%, and 2% of all ships, respectively. The bounding box areas of small ships, medium ships, and large ships account for more than 0~0.16%, 0.16~1.5%, and 1.5% of SAR images, respectively. Therefore, ships are sparsely distributed in SAR images.

The papers that used HRSID and their performance are shown in Table 9.

### 4.3.6. LS-SSDD-v1.0

Zhang Xiaoling [101] constructed the SAR ship detection dataset LS-SSDD-v1.0 with a large scene and small ships. The dataset consists of 15 pieces with a size of 24,000 × 16,000 pixels Sentinel-1 SAR images. Each image is directly divided into 600 sub-images with 800 × 800 pixels. The dataset contains 6015 ships. LS-SSDD-v1 can support researchers to flexibly apply the dataset. The optical information provided in Google Earth software and ship information provided by AIS is used for the annotation of LS-SSDD-v1.0. The coastline of the imaging area in the dataset is relatively complex, the land area is smaller than the ocean area, and the ships in the inland river are densely distributed. The dataset has the following characteristics: contains large scenes, focus on the small ships, rich pure background, etc. It also provides a large number of performance benchmarks of detection algorithms on datasets.

The papers that used LS-SSDD-v1.0 and their performance are shown in Table 10.

### 4.3.7. SRSDD-v1.0

The original images of SRSDD-v1.0 are from Gaofen-3 [177]. It contains 30 panoramic SAR images of port areas. It is annotated with an oriented bounding box. Optical images (Google Earth or GF-2) are used to assist the annotation. The image size is set to 1024 × 1024. The annotation format is the same as DOTA. The coordinates of the four corners of the box, the category, and whether it is difficult to identify are given in annotation files.

It contains 666 images. A total of 420 images with 2275 ships include the land cover. A total of 246 images with 609 ships only contain the sea in the background. It has six categories: ore-oil ships (166), bulk cargo ships (2053), fishing boats (288), law enforcement ships (25), dredger ships (263), and container ships (89). The dataset has a certain data imbalance problem.

### 4.3.8. RSDD-SAR

The RSDD-SAR dataset consists of 84 scenes of Gaofen-3 and 41 scenes of TerraSAR-X. RSDD-SAR has 7000 images, including 10,263 ships, of which 5000 are randomly selected as the training set and the other 2000 as the testing set. By analyzing the distribution of ship angle and aspect ratio in the dataset, it can be found that the angle of ships in the dataset is evenly distributed between 0° and 180°, and the aspect ratio is concentrated between two

and six. It indicates that the dataset has the characteristics of arbitrary rotation direction and a large aspect ratio. The dataset has the characteristics of a high proportion of small ships, which can be used to verify the performance of a small ship detection algorithm. The RSDD-SAR dataset contains vast sea areas, ports, docks, waterways, and other scenes with different resolutions, which are suitable for practical applications.

### 4.4. Two-Stage Detectors

The deep learning-based object detection algorithm can be divided into single-stage detectors and two-stage detectors. The single-stage detectors use a full convolution network to classify and regress these anchor boxes once to obtain the detection results. The two-stage detectors use a CNN to classify and regress these anchor boxes twice to obtain the detection results. The principles of single-stage and two-stage detection algorithms are shown in Figure 6.

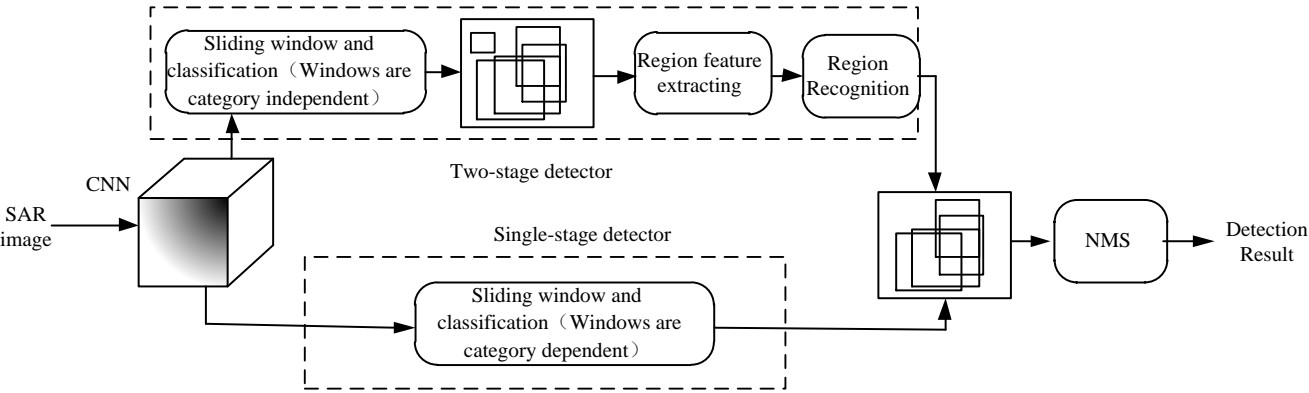

**Figure 6.** The principle of single-stage and two-stage detectors.

Classical two-stage detectors are Faster R-CNN, R-FCN (fully convolutional network) [193], feature pyramid networks (FPN) [194], Cascade R-CNN [195], Mask R-CNN [196], and so on [197]. Faster R-CNN is the foundation work, and most of the two-stage detectors are improved based on it.

Among the 177 papers, most of the papers are improved from the following aspects: backbone network, region proposal network (RPN), anchor box, loss function, and non-maximum suppressing (NMS). They are shown in Figure 7. Compared with computer vision, the research in this field lags behind, and other more advanced two-stage detection algorithms have not been used here.

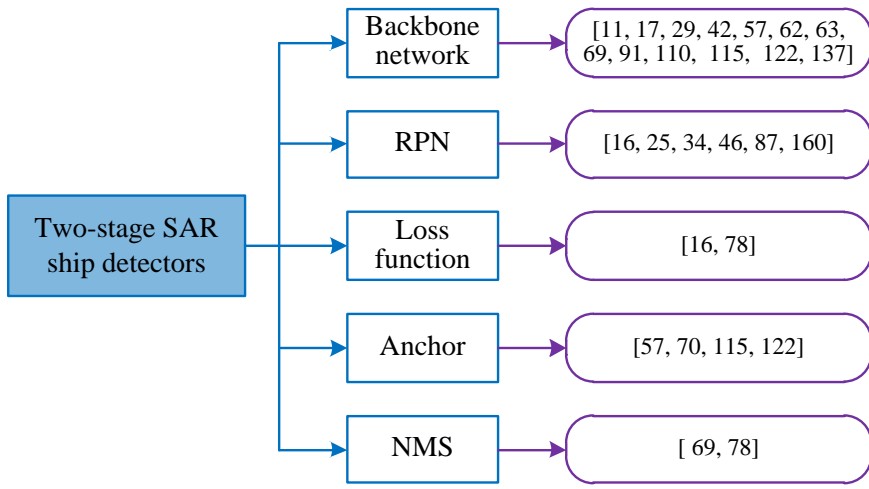

**Figure 7.** The two-stage SAR ship detectors.

### 4.4.1. Backbone Network

There are three main directions in the improvement of the backbone network, namely FPN, feature fusion, and attention.

FPN produces a feature pyramid structure that combines low-resolution, which has strong semantic features with high-resolution, which has weak semantic features. It includes a bottom-up channel, a top-down channel, and a skipping connection. It predicts independently at all levels, which only brings minimal additional calculation and storage consumption. It improves the detection result of small-size ships and thus is widely used. A lot of work has been completed to improve FPN in the computer vision field, such as ASFF [198], NAS-FPN [199], and BiFPN [200].

There are six papers [42,62,63,69,91,110] adopted and improved FPN in this field. Cui et al. [42] proposed a DAPN (dense attention pyramid network) structure. It densely connects the convolution block attention module from the top to the bottom of the pyramid network. By this, rich features including resolution and semantic information are extracted for multi-scale ship detection. Li et al. [62] used a convolution block attention module (CBAM) [201] to control the degree of upper- and lower-level feature fusion in FPN. Liu et al. [63] proposed a scale-transferrable pyramid network. It densely connects each feature map from top-to-down using scale-transfer layer. It can expand the resolution of feature maps, which is helpful for detection. Wei et al. [69] adopted a parallel high-resolution feature pyramid network to make full use of the feature mapping of high-resolution and low-resolution convolution for SAR ship detection. Zhao et al. [91] adopted receptive fields block and convolutional block attention module to build a top-down fine-grained feature pyramid. It can capture features of ships with large aspect ratios and enhance local features with their global dependences. Hu et al. [110] used a dense connection to a feature pyramid network, in which the shallow features and deep features are processed differently. It considered the differences between different levels.

There are three papers [11,57,115] that improved the backbone network through feature fusion. Li et al. [11] fused the feature maps from convolutional layer 3 to layer 5. The fusion includes the normalization and $1 \times 1$ convolution. Normalizing each RoI (region of interest) pooling tensor can reduce the scale differences between the following layers. It can prevent the 'larger' features from dominating the 'smaller' ones and make the algorithm more robust. This modification stabilizes the system and increases the accuracy. Yue et al. [57] fused the semantically strong features with the low-level high-resolution features, which is helpful for reducing false alarms. Li et al. [115] presented a jump connection structure to extract the features of each scale target in the SAR image. It can improve the ability of recognition and localization.

There are five papers [17,29,62,122,137] that improve the backbone network through the attention module (SENet). It squeezes the feature map along the space and the channel direction, which can explicitly model the interdependence between feature channels, and then automatically obtain the importance of each feature channel through learning. It can improve the useful features and suppress the features that are not useful for the current task according to the importance.

### 4.4.2. RPN

Another direction is improving the RPN module of Faster R-CNN. Paper [16,25,34,46,87] did not use a single feature map to generate proposals but generated proposals from each fused feature map. Liu et al. [36] designed a scale-independent proposal generation module, which extracts the features such as edge, super-pixel, and strong scattering component from SAR image to obtain ship proposals, and sorts whether the proposals contain ships from the integrity and tightness of the contour. In paper [160], candidate proposals are extracted from the original SAR image and the denoised SAR image, respectively, and then combined to reduce the impact of noise in the SAR image on ship detection. They can improve the performance of multi-size ships to some extent.

### 4.4.3. Loss Function

Faster R-CNN forces the ratio of positive and negative samples to 1:3 to solve the problem of unbalanced positive and negative proposals. Similar work in the field of computer vision includes focal loss, OHEM (online hard example mining) [202], GHM (gradient harmonizing mechanism) [203], and Libra R-CNN [204]. The paper [16,78], respectively, adopted focal loss to increase the weight of hard negative samples and reduce the weight of simple samples, so as to avoid the problem that a large number of simple samples cover a small number of hard negative samples in the training process.

### 4.4.4. Anchor and NMS

Faster R-CNN uses three scales and three aspect ratios, producing a total of $60 \times 60 \times 9$ anchor boxes. However, the ship size in the SAR image is extremely small and sparse. There will be a waste in using dense anchor boxes for ship detection in SAR images. Yue et al. [57] and Wang et al. [122] set the parameters of the anchor box based on the analysis of the actual size and distribution of the ship, mainly reducing the size of the anchor box and selecting the appropriate shape. Chen et al. [70] and Li et al. [115] used K-means to obtain the distribution of the ship size, so as to obtain the appropriate anchor box and reduce the difficulty of learning.

Wei et al. [69] and Wang et al. [78] used soft NMS [205] to replace NMS. Soft NMS improves the discrimination process of IoU and threshold in the cycle process and uses weights to attenuate scores to avoid accuracy loss.

### 4.4.5. Others

ISASDNet (instance segmentation assisted ship detection network) was proposed based on Mask R-CNN in the paper [163]. It has two branches: detection and segmentation. The two branches output interaction to improve the detection results. Gui et al. [34] proposed a lightweight detection head with a large separable convolution kernel and position-sensitive pooling, which improves the detection speed.

### 4.5. Single-Stage Detectors

The two-stage detectors generate a candidate box first and then identify and regress the candidate box, which is quite different from the principle of human eyes. The single-stage detectors only need to look at the picture once and can predict what the object is and where the object is. It is similar to the human eyes. In addition, they are quite faster than two-stage detectors.

Classical single-stage detectors are YOLO, SSD, RetinaNet [206], and CornerNet [207]. YOLO and SSD are the two most popular single-stage detection algorithms, and most of the subsequent single-stage works are based on them.

The single-stage ship detectors in SAR images are shown in Figure 8.

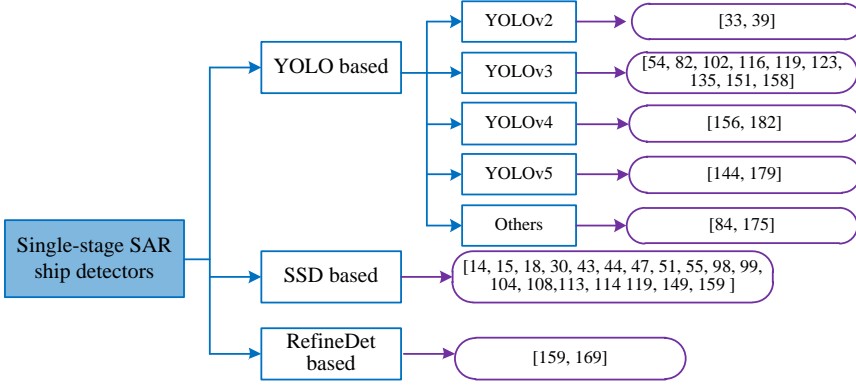

**Figure 8.** The single-stage SAR ship detectors.

### 4.5.1. YOLO and SSD Series in Computer Vision

YOLOv1 [186] regards object detection as a regression problem, and it outputs the spatially separated bounding box and related class probability simultaneously. A neural network can predict the bounding box and class probability from the image in one forward calculation. The speed is very fast, but it has an inaccurate location prediction, and the recall is low. YOLOv2 [208] uses the multi-scale training method. It predicts the offset rather than the parameter itself. The offset value is slightly smaller, which can increase the accuracy of prediction. It uses an anchor mechanism to obtain anchor box parameters by clustering the object size in the dataset. The backbone network adopts DarkNet-19. Although the detection head has changed from $7 \times 7$ to $13 \times 13$, the detection result of a small object is still poor. The YOLOv3 [209] detection head includes three branches: $13 \times 13$, $26 \times 26$, and $52 \times 52$, which can take into account large, medium, and small objects and make the location prediction more accurate. The anchor mechanism of YOLOv3 is the same as that of YOLOv2. YOLOv4 [210] uses two anchors for one ground truth, while YOLOv3 uses only one anchor for one ground truth. With this, the problem of imbalance between positive and negative samples is alleviated. CIoU loss is adopted to solve the problems of MSE (mean squared error) loss, IoU loss, GIoU, and DIoU [211–214]. YOLOv4 also uses several techniques to achieve state-of-the-art results. YOLOv5 adopts adaptive anchors and uses the network to learn anchor parameters. Its detection head is the same as YOLOv3 and YOLOv4. It is slightly weaker than YOLOv4 in performance, but much faster than YOLOv4, and has strong advantages in the rapid deployment of the model.

SSD detection algorithm combines the regression idea with the anchor box (default frame) mechanism. It eliminates the candidate region generation and subsequent pixel or feature resampling stage (RoI pooling) in the two-stage algorithm. It encapsulates all calculations in one network, making it easy to train and very fast. RFBNet [215] and M2Det [216] are two successors of SSD. They use receptive fields and multi-level feature pyramid networks to improve the classical SSD, respectively.

Single-stage SAR ship detection algorithms can be divided into three categories: SAR image ship detection based on the YOLO series, SAR ship detection based on the SSD series, and other algorithms.

### 4.5.2. SAR Ship Detection Based on YOLO Series

YOLO series are widely used in this field. The improvements mainly focus on lightweight backbone network designing, multi-layer feature fusion, anchor box generation, multi-feature map prediction, loss function, etc.

YOLOv2. Deng et al. [33] and Chang et al. [39] adopted YOLOv2 to detect ships in SAR images. Paper [39] proposed YOLOv2-reduced which reduces some layers of YOLOv2. YOLOv2-reduced has an AP of 89.76% with 10.937 ms and 44.72 BFLOPS compared with YOLOv2, which has an AP of 90.05% with 25.767 ms and 50.17 BFLOPS.

YOLOv3. Zhang et al. [82] accelerated the original YOLOv3 by using DarkNet-19 as the backbone network. Additionally, it reduces the repeated YOLOv3-scale1, YOLOv3-scale2, and YOLOv3-scale3. Zhu et al. [116], Chaudhary et al. [123], and Jiang et al. [135] used the classical YOLOv3 with some techniques to detect ships in SAR images. Wang et al. [119] proposed SSS-YOLO which redesigned the feature extractor network to enhance the spatial and semantic information of small ships. It adopts a PAFN (path argumentation fusion network) to fuse different features in a top-down and bottom-up manner. SSS-YOLO has a better performance for small ships in SAR images. Hong et al. [158] improved the performance of YOLOv3 with some techniques. The improved clustering algorithm K-means++ generates an anchor box, which improves the performance of YOLOv3 for multi-scale ships. The Gaussian parameter for ship detection is introduced to add an uncertainty estimator for the positioning of the bounding box. Four anchor boxes are assigned to each detection scale instead of three in YOLOv3. Zhang et al. [40] used the idea of the YOLO algorithm, the input image meshes, and the depth separable convolution is used to improve the detection speed. MobileNet is used as the feature extractor to detect

ships under three scales: 13 × 13, 26 × 26, and 52 × 52. The size of the anchor box can be obtained by the K-means algorithm. D-CNN-13 has a big receptive field with anchor box widths and heights of (9, 11), (11, 22) (14, 26). D-CNN-26 has a medium receptive field with anchor box widths and heights of (16, 40), (17, 12) (27, 57). D-CNN-52 has a small receptive field with anchor box widths and heights of (28, 17), (57, 28) (69, 72). Zhang et al. [54] used depth convolution and point convolution to replace the traditional convolution neural network, and adopt a multi-scale detection mechanism, concatenation mechanism, and anchor box mechanism to improve the detection speed. The detection network is composed of three parts, which means that it can detect an input SAR image under three different scales (5 × 5, 10 × 10, and 20 × 20), and then obtain the final ship detection results. It has nine anchor boxes for three detection scales, so it can detect up to nine ships in the same grid cell. Zhou et al. [102] designed a CNN named LiraNet, which has low complexity, few parameters, and a strong feature representation ability. LiraNet combines the idea of dense connections, residual connections, and group convolution, and it includes stem blocks and extractor modules. The network is the feature extractor of Libra-YOLO. Lira-YOLO has only 2.980 Bflops, and the parameter quantity is only 4.3 MB. It has good accuracy with less memory and computational cost compared with tiny-YOLOv3. In [151], DarkNet-53 with the residual unit is used as the backbone to extract features, and a top-down pyramid structure is added for multi-scale feature fusion. Soft NMS, mix-up, mosaic data augmentation, multi-scale training, and hybrid optimization are used to boost the performance. The 13 × 13, 26 × 26, 52 × 52 feature maps with the large, medium, and small receptive fields are responsible for large, medium, and small ships, respectively. The model is trained from scratch to avoid the learning objective bias of pre-training. The detection speed is fast, about 72 frames per second.

YOLOv4. Ma et al. [156] proposed YOLOv4-light, which is tailored to reduce the model size, detection time, number of computational parameters, and memory consumption. The three-channel images are used for compensating for the loss of accuracy. Liu et al. [181] proposed a detection method based on YOLOv4-Lit [217], whose backbone is MobileNetv2. A receptive field block is used for multi-scale target detection. It has an AP of 95.03% with 47.16 FPS and 49.34 M model size.

YOLOv5. Tang et al. [144] proposed N-YOLO based on YOLOv5. N-YOLO adopts a noise level classifier to classify the noise level of SAR images. SAR ship potential area extraction module is used to extract the complete region of potential ships. Zhou et al. [179] proposed a multi-scale ship detection network based on YOLOv5. It has the cross-stage partial network to improve feature representation capability, and the feature pyramid network with fusion coefficients module to fuse feature maps adaptively. It has a good tradeoff between model size and inference time.

Others. Zhang et al. [84] proposed ShipDeNet-20. It has only 20 convolution layers, and the model size is smaller than 1 MB, which is lighter than the other state-of-the-art detectors. ShipDeNet-20 is based on YOLO and is trained from scratch. Feature fusion module, feature enhance module, and scale share feature pyramid module are proposed to make up the accuracy loss of the raw ShipDeNet-20. It has a good tradeoff between accuracy and speed. Zhu et al. [175] proposed DB-YOLO. It is composed of a feature extraction network, duplicate bilateral feature pyramid network, and detection network. The single-stage network can meet the requirements of real-time detection, and it uses cross-stage partial to reduce redundant parameters. A duplicate bilateral feature pyramid network can enhance the fusion of semantic and spatial information. It alleviates the problem of small ship detection. CIoU loss is used as the loss function, as it has a faster convergence speed and better performance.

### 4.5.3. SAR Ship Detection Based on SSD Series

Wang et al. [14,18] directly used SSD and do not improve it. Papers [51,98,108] are the detection algorithms trained from scratch based on SSD. Most of the other papers improve the backbone network of SSD to make the model have a stronger feature extraction ability.

Chen et al. [15] adopted a two-stage regression network based on SSD to improve the performance of small ships, namely R2RN (robust two-stage regression network). R2RN connected an anchor modified module and object detection module to inherit the essence of the feature pyramid. Ma et al. [30] proposed an SSD model with multi-resolution input, which can extract richer features. Papers [43,44] applied the attention mechanism to SSD and design a new loss function based on GIoU. Li et al. [47] analyzed the reasons for the low detection accuracy of small and medium-sized ships in SSD and puts forward improvement strategies. Firstly, the anchor box optimization method based on K-means clustering is adopted to improve the matching performance of the anchor box. Secondly, a feature fusion method based on deconvolution is proposed to improve the representation ability of the underlying feature map. Chen et al. [55] adopted the attention mechanism and multi-level features to improve the feature extraction ability of the backbone network. Han et al. [99] used deconvolution to enhance the representation of small ships in the pyramid and improved the detection accuracy of SSD. Zhang et al. [113] token the original SAR image and saliency map as the input and fused the fusion of their features to reduce the computational complexity and network parameters. Chen et al. [114] proposed SSDv2, which adds a deconvolution module and prediction module on the basis of SSD to improve the detection accuracy. Jin et al. [149] improved SSD by feature fusion and squeeze-excitation module.

Sun et al. [162] proposed SANet (semantic attention-based network). It combines semantic attention, focal loss, label, and anchor assigning to improve the performance without increasing computation. Papers [104,159] adopted M2Det to detect ships in SAR images.

### 4.5.4. Others

RefineDet adopts a two-step cascade regression strategy to predict the position and size of objects. It can make the single-stage detectors obtain the accuracy of the two-stage detector without increasing computation. It is widely used in computer vision. Zhu et al. [159] adopted RefineDet to detect ships in SAR Images, which achieve an AP of 98.4%. In [169], GHM was used as the loss function of RefineDet, so that the network can make full use of all examples, and adaptively increase the weight of difficult cases. A multi-scale feature attention module is added to the network to highlight important information and suppress the interference caused by clutter. It achieves 96.61% precision on AIR-SARShip-1.0.

### 4.6. Anchor Free Detectors

### 4.6.1. Development of Anchor Free Detection Algorithm in Computer Vision

The anchor box is the key to the success of Faster R-CNN and SSD. The backbone network extracts features from the input image to obtain the feature map, and each pixel on the map is the anchor point. Taking each anchor point as the central point and artificially setting different scales and aspect ratios, multiple anchor boxes can be obtained. Anchor box has the following two advantages: firstly, it can generate dense candidate boxes, which is convenient for the network to classify and regress the targets. Secondly, it can improve the recall ability and is suitable for small target detection.

However, the anchor box needs to be designed manually by experience, which has the following defects: firstly, hyper-parameters need to be set, such as the number, size, aspect ratio, IoU threshold, etc. Secondly, in order to match the ground box, a large number of anchor boxes need to be generated, which are computationally intensive. Thirdly, most of them are invalid, which will lead to an imbalance between positive and negative samples. Fourthly, it is necessary to adjust the anchor box according to the size and shape distribution of the dataset.

The anchor-free detector opens up another idea by eliminating the predefined anchor box. It can directly predict several key points of the target from the feature map. For example, CornerNet, ExtremeNet [218], CenterNet [219], Objects as Points [220], FCOS (fully convolutional one-stage) [221] and FoveaBox [222].

The anchor-free detectors can avoid various problems and has great application potential in SAR ship detection. For example, due to the small size and sparse distribution of ships, most of the candidate anchor boxes are invalid negative samples. The anchor-free detectors can neglect the invalid anchors and reduce the amount of the predicted boxes, thus improving the accuracy and speed simultaneously. The anchor-free ship detectors in SAR images are shown in Figure 9.

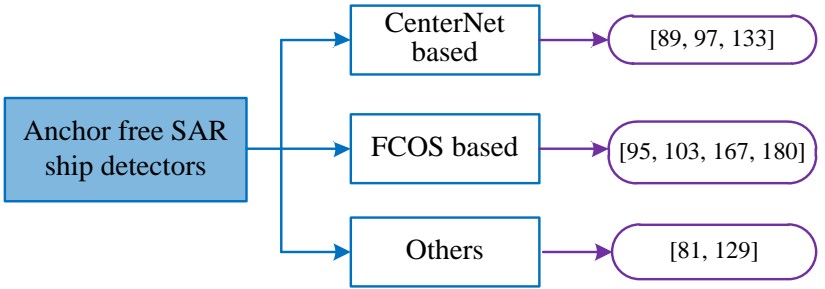

**Figure 9.** The anchor-free SAR ship detectors.

### 4.6.2. Development of Anchor-Free SAR Ship Detection Algorithm

Mao et al. [81] proposed a simplified U-Net [223] based anchor-free detector in SAR images. It includes ship bounding boxes regression network and score map regression network. The former is expected to be regressed based on each pixel in the input image. The latter is designed to predict a 2D probability distribution in which each score at each position indicates the likelihood of the current position in the center of any ship. Cui et al. [89] proposed a CenterNet (objects as points) based SAR ship detector. It predicts the center point of the target through key point estimation and uses the image information of the center point to obtain the size and position of the ship. There is no need to set anchors in advance and NMS is not needed, which greatly reduces the number of network parameters and calculations. Anchor mismatching of small ships is also reduced. Spatial shuffle-group enhance attention modules are used to extract features with more semantic information. Fu et al. [95] proposed an attention-guided balanced pyramid based on FCOS to improve the performance of small ships. Zhou et al. [97] proposed an anchor-free detector with dense attention feature aggregation. A lightweight feature extractor and dense attention feature aggregation are used to extract multi-scale features. A center-point-based ship predictor is used to regress the centers and sizes. There is no pre-set anchor and NMS, and thus the computational efficiency is high. Mao et al. [103] proposed a lightweight named ResSARNet with only 0.69 M parameters, and improved FCOS in four aspects: center-ness in bounding box regression branch, not in classification regression branch, center sampling, GIoU loss, and adaptive training sample selection. The network only needs 1.17 M parameters and can achieve 61.5% AP and 70.9% AR. An et al. [129] designed an anchor-free rotatable detector. It designs center point-scale and angle prediction to convert the conventional rotatable prior box mechanism into the center point-scale and angle prediction. The training procedure includes positive sample selection, feature encoding, and loss function designing. Wang et al. [133] proposed a CenterNet-based detector in SAR images. The spatial group-wise enhanced attention module is used to extract more semantic features.

Sun et al. [167] proposed category-position FCOS. The category-position module is used to optimize the position regression branch in the FCOS network. The classification and regression branches are redesigned to alleviate the imbalance between positive and negative samples during training. Zhu et al. [180] adopted FCOS as the base model to reduce the effect of anchors. A new sample definition method is used to replace the IoU threshold according to the differences between SAR images and natural images. The same resolution feature convolution module, multi-resolution feature fusion module, and feature pyramid module are used to extract features. The focal loss and CIoU are used to improve the performance further.

In all, researchers in SAR ship detection realize the benefit of anchor-free detectors. Additionally, more and more papers are appearing in this field. However, there is still a problem: the innovation is relatively weak and some of the existing achievements of computer vision are not used in this field.

*4.7. Detectors Trained from Scratch*

At present, most SAR image detector backbone needs to pre-train on the classification dataset of natural images, and then fine-tune on the ship detection dataset of SAR images (for example SSDD). This transfer learning can make the detection algorithm initialize better and make up for the problem of insufficient samples. However, there will be the following problems: firstly, there is learning bias. The loss function and category distribution between classification and detection are contradictory in essence. The models trained on classification are not fit for detection. Secondly, most backbone networks will produce a high receptive field through multiple down sampling in the latter layers, which is good for the classification but is harmful for the location. Thirdly, the pre-trained backbone networks are redundant and cannot be modified, which hinders the researchers to design CNN flexibly according to their needs.

In order to solve the problems of transfer learning, algorithms trained from scratch are proposed in computer vision, for example, DSOD (deeply supervised object detectors), DetNet, ScratchDet, and so on [224–227].

The main idea of DSOD and GRP-DSOD to realize training from scratch is by designing the backbone and front-end network elaborately [224]. DetNet [225] retains a large scale in the last few layers, which can have more location information. ScratchDet [226] proposes to adopt the strategy of batch normalization in each layer and increase the learning rate, which can make the detection algorithm more robust and converge faster. Paper [227] replaced the original BN (batch normalization) with group normalization (GN) and asynchronous BN, so as to make the parameters of gradient normalization more accurate. This then made the descending direction of gradient more accurate, so as to accelerate convergence and improve the accuracy.

The model trained from scratch not only has high accuracy but also greatly reduces the size and amount of calculation of the model. Due to the above advantages, it is also used in SAR ship detection.

Most detectors that are trained from scratch in this field have well-designed networks. They are shown in Figure 10.

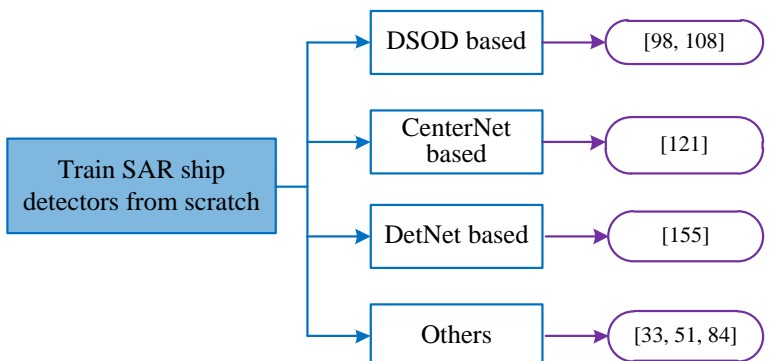

**Figure 10.** The detectors trained from scratch in SAR images.

Deng et al. [33] designed a dense backbone network composed of multiple dense blocks. The front layer can receive additional supervision from the objective function through dense connections, which makes the training easier, and adopts the feature reuse strategy to make the parameter highly efficient. Zhang et al. [51] designed a lightweight detection algorithm that can be trained from scratch, which can reduce the training and testing time without reducing the accuracy. It adopts the modules of semantic aggregation and feature reusing to improve the performance of multi-scale ships. Zhang et al. [84]

proposed a lightweight detection network ShipDeNet-20. It is designed with fewer layers and convolution kernels and depth separable convolution. It also adopts a feature fusion module, feature enhancement module, and proportionally shared feature pyramid module to improve detection accuracy. Han et al. [98] integrated the lightweight asymmetric square convolution block into SSD to realize training from scratch, and its accuracy and speed are better than the classical DSOD. Han et al. [100] proposed a parallel convolution block of multi-scale kernel and feature reusing convolution module to enhance feature representation and reduce information loss. Han et al. [108] designed two kinds of asymmetric convolution blocks: asymmetric and square convolution feature aggregation block, and asymmetric and square convolution feature fusion block. They replace all $3 \times 3$ convolution layers, which are embedded into the classic DSOD to achieve a better result of the training from scratch. Guo et al. [121] proposed an effective and stable single-stage algorithm that is trained from scratch, namely CenterNet++. The model mainly includes three modules: feature matching module, feature pyramid fusion module, and head enhancement module. Zhao et al. [155] used DetNet as the backbone network to realize training from scratch. It uses superposition convolution instead of down sampling to solve the problem of small ship detection and adopts a feature reusing strategy to improve parameter efficiency.

Compared with other directions, fewer researchers in SAR ship detection realize the benefit of training from scratch. Additionally, the papers using training from scratch techniques in this field are not advanced enough. We should adopt more advanced techniques in computer vision in this direction. In all, the detectors trained from scratch are not used to their full extent in this field. Some good conclusions in papers [226,227] should be considered and applied here.

### 4.8. Detectors with Oriented Bounding Box

The oriented bounding box was originally used in scene text detection. In addition, a large number of achievements have emerged, such as SegLink, RRPN (rotation region proposal network), TextBoxes, TextBoxes++, R2CNN (rotational region convolutional neural network), and so on [228–232]. The ships in remote sensing images also have multi-directional characteristics. The conventional vertical rectangular bounding box often cannot accurately surround the target. With the improvement of ship detection accuracy, the use of oriented bounding boxes to realize multi-directional ship detection has become a research hotspot [233–238]. DOTA (dataset for object detection in aerial images) is a commonly used aerial image target detection dataset in this field, which can be used to develop and evaluate the performance of detection algorithms. Similarly, there are many detection algorithms based on oriented bounding boxes in SAR images, which will be introduced here. At present, the dataset that can be used to train and test the oriented bounding box algorithm are SSDD+, RSDD-SAR, and SRSDD-V1 0, the details have been introduced earlier. The oriented bounding box detectors in SAR images are shown in Figure 11.

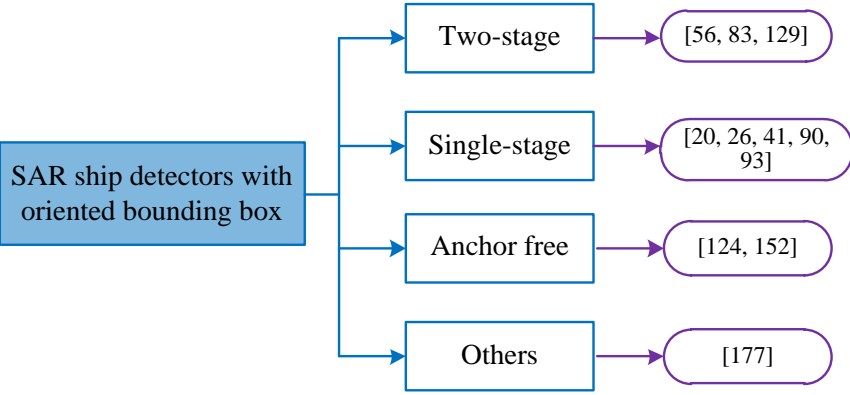

**Figure 11.** The SAR ship detectors with oriented bounding box.

Two-stage. Chen et al. [56] proposed a multi-scale adaptive recalibration network to detect multi-scale and arbitrarily oriented ships. It can learn the angle information of ships. The anchors, NMS, and loss function are also redesigned to fit the large aspect ratio and arbitrary directionality of ships in SAR images. Pan et al. [83] proposed a multi-stage rotational region-based network (MSR2N) to solve the problem of redundancy regions. MSR2N includes FPN, RRPN, and a multi-stage rotational detection network. It is more suitable and robust for SAR ship detection. An et al. [129] adopted an oriented detector as the based model to solve the problem that conventional CNN models have too many parameters, which increases the difficulty of transfer learning between different tasks.

Single-stage. Wang et al. [20] proposed a SAR ship detector with an oriented bounding box based on SSD. The detector can predict the class, location, and angle information of ships. The semantic aggregation module is used to capture abundant location and semantic information. The attention module is used to adaptively select meaningful features and neglect weak ones. Multi-orientation anchors, angular regression, and the loss function are used to fit the oriented bounding box. Liu et al. [26] adopted DR-Box [239] to detect ships in SAR images. DR-Box is specially designed to detect targets in any direction in remote sensing images. It can effectively reduce the interference of background pixels and locate the target more accurately. An et al. [41] proposed DR-Box-v2 to detect ships in SAR images. A multi-layer anchor box generation strategy for detecting small ships is proposed. A modified encoding scheme is proposed to estimate the position and orientation precisely. Focal loss and hard negative mining are also used to balance the positives and negatives. Yang et al. [90] regarded a rotatable bounding box detector as the base model to solve the problem of negative sample intra-class imbalance in the training stage. Chen et al. [93] proposed a rotated refined feature alignment detector to fit ships with large aspect ratios, arbitrary orientations, and dense distribution properties. A lightweight attention module, modified anchor mechanism, and feature-guided alignment module are proposed to boost the performance of the oriented detector.

Anchor free. Yang et al. [124] proposed R-RetinaNet to beat DRBox-v1, DRBox-v2, and MSR2N (multi-stage rotational region-based network) in this field. R-RetinaNet used a scale calibration method to align the scale distribution. Task-wise attention feature pyramid network is used to alleviate the contradiction of classification and localization. The adaptive IoU threshold training method is used to correct the unbalanced problem. He et al. [152] proposed a method to solve the problem of boundary discontinuity problem in oriented bounding box detectors by learning polar encodings. The encoding scheme uses a group of vectors pointing from the center of the ship to the boundary points to represent an oriented bounding box.

Others. Ding et al. [177] released the SRSDD-v1.0 dataset, which is used for oriented bounding box detectors. The details of the dataset have been described above. They present the performance of several advanced oriented bounding box detection algorithms on the dataset.

Summary. With the emergence of several datasets with oriented bounding boxes, ship detectors in SAR images based on oriented bounding boxes are becoming more and more advanced. However, it is not enough compared with DOTA. Some efforts should be taken in this direction.

### 4.9. Multi-Scale Ship Detectors

In MS COCO, the proportions of the small, medium, and large size objects are 41.43%, 34.32%, and 24.24%, respectively. However, in SAR images, the proportion of small-size ships is extremely high. For example, the proportion of small, medium, and large ships in the RSDD-SAR dataset are 81.175%, 18.776%, and 0.049%, respectively. In LS-SSDD-v1.0, the proportions are 99.80%, 0.20%, and 0.00%, respectively. Therefore, this field needs to focus on the problem of multi-scale ship detection, especially the small ships.

Although CNN has developed rapidly in computer vision, it has poor performance on small-size object detection. In order to improve the adaptability to multi-scale ships,

computer vision often fuses low-level and high-level features (such as FPN), increases the receptive field and improves the anchor box generation and matching strategy.

SAR ship detection also uses the above methods to improve the performance of multi-scale ship detection. The multi-scale ship detectors in SAR images are shown in Figure 12.

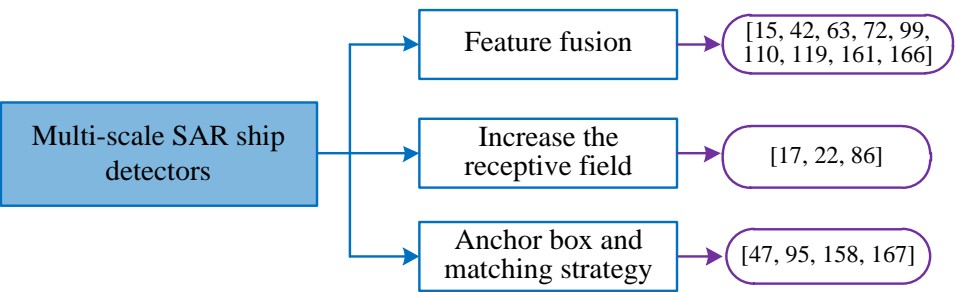

**Figure 12.** The multi-scale SAR ship detectors.

Feature fusion. Chen et al. [15] proposed a densely connected multi-scale neural network to solve the problem of multi-scale SAR ship detection. It closely connects each feature map with other feature maps from top-to-bottom and generates proposals from each fused feature map. Cui et al. [42] proposed a dense attention pyramid network, which closely connects the convolutional attention module from the top to the bottom of the pyramid network to each feature map, so as to extract rich features containing location information and semantic information for adapting to multi-scale ships. Liu et al. [63] proposed a scale transferable pyramid network to adapt to the detection of multi-scale ships. It constructs a feature pyramid network through horizontal connection and uses a scale transfer layer to closely connect each feature graph from top to bottom. A horizontal connection introduces more semantic information, and a dense scale transfer connection can expand the resolution of the feature map. Jin et al. [72] combined all feature maps from top to bottom to make use of contextual semantic information at all scales and uses extended convolution to increase the receptive field exponentially. Han et al. [99] used deconvolution to enhance the feature representation of small and medium-sized ships in FPN, so as to improve the detection accuracy of SSD. Hu et al. [110] proposed a dense feature pyramid network, which processes shallow features and deep features differently. Compared with traditional FPN, it has stronger adaptability to multi-scale ships. Wang et al. [119] proposed a path argumentation fusion network to fuse different feature maps. It uses bottom-up and top-down methods to fuse more location information and semantic information. Hu et al. [161] proposed a two-way revolution network based on a bidirectional convolution structure, which can effectively process shallow and deep feature information and avoid the loss of small ship information. Zhang et al. [166] proposed a quad feature pyramid network to detect multi-scale ships. It includes deformable convolutional FPN, a content-aware feature reassembly FPN, a path aggregation space attention FPN, and a Balance Scale Global Attention FPN.

Increase the receptive field. Deng et al. [17] designed a feature extractor with multiple receptive fields through ReLU and inception modules. It generates candidate regions in multiple middle layers to match ships of different scales and fuses multiple feature maps so that small-scale ships have a stronger response. Zhao et al. [22] proposed a coupled CNN to detect small-scale ships. It includes a network that generates candidate areas from multiple receptive fields and improves the recognition accuracy by using the context information of each candidate box. Dai et al. [86] did not use a single feature map but fused the feature map in a bottom-up and top-down manner, and generated candidate boxes from each fused feature map.

Anchor box generation and matching strategy. Li et al. [47] first analyzed the reasons for the low detection accuracy of small and medium-sized ships in SSD and made some improvements. The anchor box optimization method based on K-means clustering solves the problem of less positive samples and more negative samples. The feature fusion

method based on deconvolution improves the representation ability of the low-level feature map and solves the weak recognition ability of the low-level feature map to small ships. Fu et al. [95] proposed a feature balance and matching network, which uses the anchor-free strategy to eliminate the influence of anchors and uses the attention-guided balance pyramid to balance multiple features at different levels semantically. It has a good performance in the detection of small-scale ships. Hong et al. [158] improved the anchor generation based on an improved K-means++ in YOLOv3. It alleviates the difficulty of multi-scale ship detection in YOLOv3 and changes the number of anchor boxes in the YOLO layer from three to four. Sun et al. [167] show that anchor-free detectors have good adaptability to small ships and have a fast speed.

Summary. Small ship detection is extremely hard but is also extremely important for some applications. That is because people hope to find targets within a long distance, and at this point, the targets must be small in size. SAR ship detection also proves this point. Although the above detection methods for small-size ships have certain effects, they are still far from enough. Innovative work needs to be continued.

*4.10. Attention Module*

The basic idea of the attention mechanism in computer vision is to make the model ignore irrelevant information and focus on key information. It can be divided into hard attention, soft attention, gaussian attention, spatial transformation, and so on. Attention can be calculated from the spatial domain, channel domain, layer domain, and mixed domain. Representative algorithms include SENet (squeeze and excitation network), SKNet (selective kernel network), CBAM (convolutional block attention module), CCNet (criss-cross attention), OCNet (object context network), DANet (dual attention network), etc. [240–244]. Transformer [245] adopted encoder–decoder architecture, which is the extreme of the attention. It abandons CNN and RNN (recurrent neural network) used in previous deep learning tasks and shows great advantages in the field of NLP (natural language processing) and CV. Swin Transformer [246] makes it compatible with image classification and object detection. It demonstrates the potential of transformer-based models as vision backbones.

Chen et al. [43,44] proposed an attention-based detector. The attention model is mainly composed of the convolution branch and mask branch. Elements in mask maps are similar to the weight of feature maps, which enhance regions of interest and suppress non-target regions. Cui et al. [89] introduced the space shuffle group enhanced attention module to CenterNet. It can extract stronger semantic features and suppress some noise at the same time, so as to reduce false positives caused by inshore and inland interference. Zhao et al. [91] combined the receptive field module and convolution block attention module to construct a top-down fine-grained feature pyramid. Wang et al. [122] designed a feature enhancement module based on a self-attention mechanism. Its spatial attention and channel attention work at the same time to highlight the target and suppress the spot to a certain extent. Wang et al. [131] embedded a soft attention module in the network to suppress the influence of noise and complex background. Zhu et al. [136] proposed a SAR ship detection method based on a hierarchical attention mechanism. The method includes a global attention module and a local attention module. Hierarchical attention strategies are proposed from the image layer and target layer, respectively. Sun et al. [162] introduced a semantic attention mechanism, which highlights the regional characteristics of ships and enhanced the classification ability of the detector. Du et al. [169] embedded the multi-scale feature attention module in the network. By applying the channel and spatial attention mechanism to the multi-scale feature map, it can highlight important information and suppress the interference caused by clutter.

CRTransSar [182] is the first to use a transformer for SAR image ship detection. It is based on Swin Transformer and shows great advantages. CRTransSar combines the global contextual information perception of transformers and the local feature representation capabilities of convolutional neural networks. It innovatively proposes a visual transformer

framework based on contextual joint-representation learning. Experiments on SSDD and SMCDD show the effectiveness of the method.

### 4.11. Real-Time Detectors

At present, deep learning-based detectors need large computation and storage resources, which hinders the application in real-time prediction. In order to solve this problem, there are a lot of acceleration ideas in the evolution of object detection algorithms. Firstly, researchers usually speed up the detection process. This idea is reflected in the evolution process of R-CNN, Fast R-CNN, Faster R-CNN, R-FCN, and Light-Head R-CNN. The above detectors share the features gradually, and the network structures become thinner and faster. Secondly, researchers usually design lightweight detection networks. The backbone network and the detection head can both be accelerated. Thirdly, researchers usually compress and accelerate CNN models. It includes lightweight neural network designing, model pruning, model quantization, and knowledge distillation [247–253].

The exploration of real-time detection algorithms in SAR ship detection can be divided into three directions, which are shown in Figure 13.

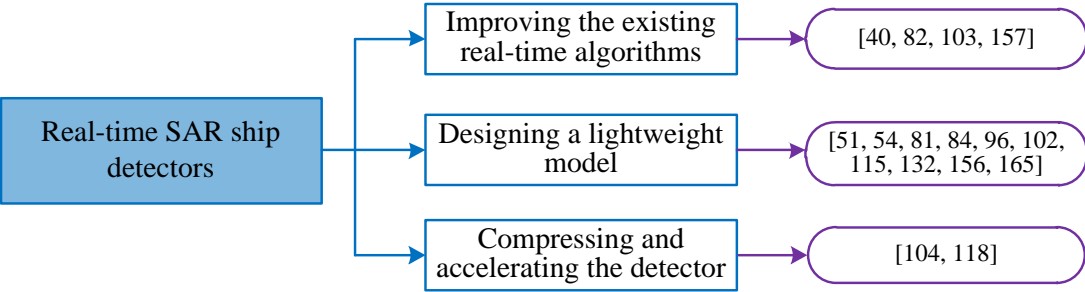

**Figure 13.** The real-time SAR ship detectors.

#### 4.11.1. Improving the Existing Real-Time Algorithms

Many improvements in this field are based on the YOLO and SSD series, because they have great advantages in running time, especially the YOLO series. Zhang et al. [40] used the idea of the YOLO algorithm and adopted depth separable convolution to accelerate the speed. MobileNet is used as the backbone network to improve the detection speed under the condition of ensuring detection accuracy. Zhang et al. [82] proposed an improved YOLOv3 (using DarkNet-19 and deleting repeated layers). It achieved 90.08% $AP_{50}$ and 68.1% AP on the SSDD dataset. Mao et al. [103] adopted the FCOS detection algorithm with ResSARNet as the backbone network, and center-ness on bounding box regression branch, center sampling, GIoU loss, and adaptive training sample selection were used. It can achieve 61.5% AP with only 1.17 M parameters. Zhong et al. [157] adopted CFAR and YOLOv4 to realize real-time ship detection on China HISEA-1 SAR images.

#### 4.11.2. Designing a Lightweight Model

Zhang et al. [51] designed a lightweight feature optimization network LFO-Net based on SSD. It can be trained from scratch and reduce the training and testing time without reducing the accuracy. The detection performance is further improved by the bidirectional feature fusion module and attention mechanism. It achieved 80.12% $AP_{50}$ with 9.28 ms testing time on SSDD. Zhang et al. [54] used multi-scale detection, cascade, and anchor box mechanism to design a lightweight network for real-time SAR ship detection. It uses depthwise and pointwise to replace the traditional convolution. It achieved 94.13% $AP_{50}$ with 9.03 ms testing time on SSDD. Mao et al. [81] used the simplified U-Net as the feature extraction network, which has only 0.47 million learnable weights, it improves the operation speed and solves the problem caused by the anchor box through the anchor-free method. It has a total of 0.93 million learnable weights, and the AP on the SSDD dataset is 68.1%. Zhang et al. [84] proposed ShipDeNet-20, which has 20 convolution layers and a

0.82 MB model size. It uses fewer layers and kernels, and depthwise separable convolution is also used. It improves the accuracy through the feature fusion module, feature enhancement module, and scale share feature pyramid module. It achieved 97.07% $AP_{50}$ with 233 FPS on SSDD. Zhang et al. [96] proposed HyperLiNet. It realizes high precision through five modules, namely multi receptive field module, divided revolution module, channel and spatial attention module, feature fusion module, and feature pyramid module. It realizes high speed through five modules, namely region-free model, small kernel, narrow channel, separate revolution, and batch normalization fusion. Zhou et al. [102] proposed a lightweight detector Lira YOLO. It combines the idea of dense connections, residual connections, and group convolution, including stem blocks and extractor modules. It achieved 85.46% $AP_{50}$ with a 4.3 MB model size. Li et al. [115] designed a lightweight network of feature relay amplification and multi-scale feature jump connection structure based on Faster R-CNN and improves the selection of anchor boxes and RoI pooling. It achieved 89.8% $AP_{50}$ and the speed increased a lot. Zhang et al. [132] proposed a lightweight detection algorithm ShipDeNet-18, which has fewer layers and fewer convolution kernels. The deep and shallow feature fusion module and a feature pyramid module are adopted to improve the detection accuracy. It achieved 93.78% $AP_{50}$ with 202 FPS. Ma et al. [156] proposed YOLOv4-tiny. It reduces the number of convolutional layers in CSPDarkNet53. It achieves 88.08% $AP_{50}$ with 12.25 ms compared with YOLOv4 with 96.32% AP and 44.21 ms. Sun et al. [165] proposed a lightweight densely connected sparsely activated detector. It can construct a lightweight backbone network, so as to achieve a balance between performance and computational complexity. It achieved 97.2% $AP_{50}$ and 61.5% AP on SSDD.

### 4.11.3. Compressing and Accelerating the Detector

Mao et al. [104] proposed a knowledge distillation-based network slimming method. YOLOv3 and Darknet-53 are pruned on filter-level to obtain lightweight models. Kullback Leibler Divergence (KLD) knowledge distillation is used to train student and teacher networks (YOLOv3@EfficientNet-B7). The model has only 15.4 M parameters, and the AP decreases by only 1%. Chen et al. [118] proposed the algorithm of Tiny-YOLO-Lite. It designs and prunes the backbone structure, strengthens the channel level sparsity, and uses knowledge partition to make up for the performance degradation caused by pruning. Tiny-YOLO-Lite reduces the size of the model, reduces the number of floating-point operations, and obtains faster accuracy.

### 4.11.4. Summary

From the above discussion, we can find that real-time ship detection is also a hot topic in SAR images. However, the above works are not enough. It is obvious that the transferred deep learning models from computer vision are abundant in this field. Researchers should do the following work to realize real-time detection. Firstly, the anchor-free and the training from scratch method should be used to design lightweight detection algorithms. Secondly, some model compressing, and accelerating techniques should be used to improve the speed further. Thirdly, the lightweight models should be transplanted to high-performance AI chips (NVIDIA Jetson TX2) to achieve the purpose of running at the edge (satellite, airplane).

### 4.12. Other Detectors

In this part, we mainly introduce weakly supervised, GAN (generative adversarial network) and data augmentation, which are shown in Figure 14.

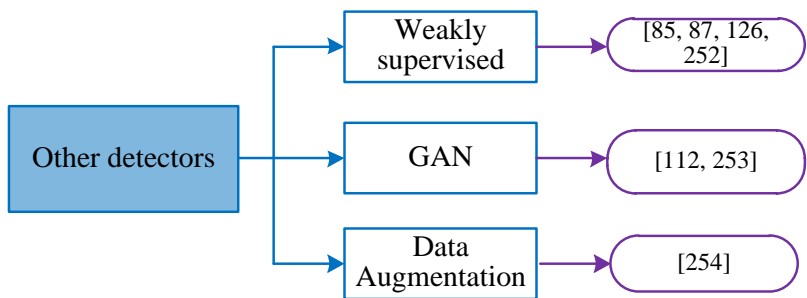

**Figure 14.** The other SAR ship detectors.

### 4.12.1. Weakly Supervised

The supervised methods, such as deep learning approaches, need substantial time and manpower to make training samples [254]. Papers [85,87,126] adopted weakly supervised to train ship detection algorithms. The model is trained by two global labels, namely, "ship" and "non-ship," and produces a ship location heatmap, ship bounding box, and pixel-level segmentation product. They can alleviate the problem of annotation partly. However, the accuracy is lower than the supervised method.

### 4.12.2. GAN

The insufficient SAR samples restrict the performance of the algorithm. Zou et al. [112] used a multi-scale Wasserstein auxiliary classifier generative adversarial network [255] to generate high-resolution SAR ship images. Then, the original dataset and the generated data are combined into a composite dataset to train the YOLOv3 network, so as to solve the problem of low detection accuracy under a small dataset. Based on the idea of generative adversarial networks, an image enhancement module driven by target features is designed. The quality of the ships in the image is improved. The experimental results verify the effectiveness of this method.

### 4.12.3. Data Augmentation

Data augmentation can expand the size of the dataset several times, so as to improve the detection accuracy [256]. The training method based on a feature mapping mask eliminates the gradient noise introduced by random clipping, so as to improve the detection performance. The SAR images with ships are generated by electromagnetic numerical analysis technology, and the sea clutter model is used to simulate the real SAR image patch containing various SAR slices, so as to improve the performance of SSD.

### *4.13. Problems*

From the 177 papers, we can see that most of the detection algorithms in this field are borrowed from computer vision. Additionally, its development is also behind the detectors in computer vision. Due to the large difference between natural image and SAR image (for example, SAR image is single-channel, ship size is small, and distribution is very sparse), some detection algorithms are not suitable for SAR ship detection. So, we should design detectors according to the real characteristics of ships in SAR images.

The 177 papers mainly use the image essences of SAR images, and the research and application of the scattering mechanism are not enough. This is one problem we should solve in the future.

At present, there are several public small datasets, but we lack a large dataset. The models trained on a small dataset face the problem of over-fitting. What we should do next is merge the small datasets into a large one, and make sure the train-test division standards, evaluation indicators, and benchmarks are clear. These works can promote the development of this field.

## 5. Future—The Direction of the Deep Learning-Based SAR Ship Detectors

### 5.1. Anchor Free Detector Deserves Special Attention

The anchor-free detection algorithm has many advantages, which have been introduced in Section 4.6. It should be emphasized that the detection algorithm without an anchor box is especially suitable for SAR images. As SAR images have sparse and small size ships, it can greatly improve the detection speed and avoid various problems in anchor box designing and matching. Therefore, the anchor-free detection algorithm needs to be paid more attention. Fortunately, researchers in this field have realized this and many research results have emerged.

### 5.2. Train Detector from Scratch Deserves More Attention

At present, there are the following generally accepted conclusions about training from scratch: firstly, pre-training accelerates the convergence speed, especially in the early stage of training. However, the training time of scratch is roughly equivalent to the total time of pre-training and fine-tuning. Secondly, if there are enough target images and computing resources, pre-training is not necessary. Thirdly, if the cost of image collection and image cleaning is considered, a general large-scale classification dataset is not an ideal choice, and collecting images on detection tasks will be a more effective approach. Fourthly, when the target task is to predict spatial positioning (such as ship detection), pre-training does not show any benefits.

Collecting images for detection and training is a solution worth considering, especially when there is a significant gap between the pre-training task and the detection task (such as ImageNet image and SAR image). Therefore, in the field of SAR ship detection, it is very necessary to combine the existing public datasets into a large dataset, so as to ensure training models from scratch.

Due to the difference between natural images and SAR images, it is very necessary to adopt training from scratch detection algorithms in this field, as it can obtain a detection algorithm with stronger adaptability to SAR images and smaller model size. However, the work at this stage is far from enough, so we need to pay more attention to the detection algorithms of training from scratch.

### 5.3. Many Other Works Need to Be Used for Oriented Bounding Box Detector

The ship in the SAR image has very changeable directionality, and the vertical bounding box cannot adapt to this scene. It is necessary to use an oriented bounding box. In an inshore scenario, a vertical bounding box is susceptible to interference from onshore buildings and other ships, affecting detection performance, while an oriented bounding box can accurately represent the ship target and reduce redundant interference. In addition, for ship targets in an offshore scenario, an oriented bounding box can obtain information such as heading and aspect ratio, which is of great significance for subsequent trajectory prediction and situation estimation tasks. The scene text detection and the aerial remote sensing image dataset DOTA have conducted in-depth research on the oriented bounding box and achieved many results. We should learn from them.

### 5.4. Small Ship Detection Is an Eternal Topic

The main reasons for the poor detection result of small-size ships are as follows: firstly, the features extracted from small-scale ships are few, and the size and receptive field of the anchor are too large for small ships. Secondly, the size of the anchor is discrete (for example, 16, 32, 64, etc.), while the size of the ship is continuous, which makes the recall rate of small-size ships low. Thirdly, the anchor of a small ship matches less with the ground truth bounding box, resulting in fewer positive samples and too many negative samples.

As the proportion of small ships in SAR images is very high, small object detection is difficult, especially in this field. So, it is an eternal topic to study how to improve the detection effect of small ships.

### 5.5. Real-Time Detection Is the Key to Application

Real-time SAR ship detection needs to start from many aspects. For example, we can design a lightweight detection network, compress and accelerate the model to improve the speed, and transplant the detection algorithm to high-performance AI chips at the edge (NVIDIA Jetson TX2). At present, most of the work in this field is focused on the first two aspects, and there is less research on the third aspect, which needs to be focused on in the future. Only by realizing this technology can we realize the real-time detection and recognition of ships on satellite or aircraft platforms.

### 5.6. Transformer Is the Future Trend

In the past two years, transformer shows great advantage in object detection compared with CNN, for example, DETR (detection transformer) [257] and Swin Transformer. DINO [258] can achieve 63.6% AP on the COCO test-dev, which surpasses CNN-based detector by a large margin. Nowadays, the hot topic of computer vision is the transformer. CRTransSar [182] is the first to use a transformer for SAR image ship detection. It shows a great advantage in accuracy (97% AP on SSDD). Although there are still some problems when a transformer is used for SAR ship detection, there is no doubt that the transformer will be the research trend in the future due to its great advantages.

### 5.7. Bridging the Gap between SAR Ship Detection and Computer Vision

Compared with the field of computer vision, the field of ship detection in SAR images is relatively small and not active enough. Therefore, it is necessary to bring this field to computer vision, and systematically learn from the rich achievements in computer vision.

We should also learn about its openness, standardized evaluation, and easily accessed codes. This work can promote this field to develop rapidly. What we should do is as follows: firstly, the existing public datasets (SSDD, SAR-Ship-Dataset, AIR-SARShip, HRSID, LS-SSDD-v1.0, SRSDD-v1.0, and RSDD-SAR) need to be combined into a large dataset, which can be called LargeSARDataset here. The modes trained on it can avoid over-fitting. Secondly, determine the training samples and testing samples. Thirdly, determine the evaluation indicators. Fourthly, release the benchmark. Fifthly, bring it into the field of computer vision. As shown in Figure 15. Through this work, we can bridge the gap between SAR ship detection and computer vision.

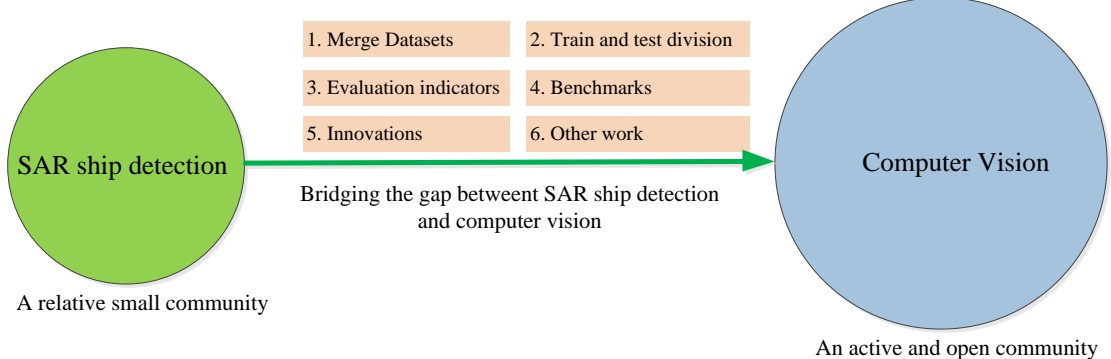

**Figure 15.** Bridging the gap between SAR ship detection and computer vision.

In addition to detection, classification and segmentation of SAR images also enter into the deep learning era [259–265]. Classification and segmentation algorithms borrowed from computer vision are extensively used in SAR images. We will review them in the future. In the process of detection, only ship and non-ship targets are considered, and the specific content of non-ship targets is not analyzed [266–268]. Some icebergs have great similarities in shape and size with ships, and the algorithms are difficult to distinguish them. So, we will study how to solve this problem in the future.

## 6. Conclusions

This paper introduces the past, present, and future of deep learning-based ship detection algorithms in SAR images.

Firstly, the history of SAR ship detection is reviewed (before SSDD was public on 1 December 2017). This part mainly introduces the detection algorithm based on CFAR and analyzes the great advantages of deep learning-based algorithms. In addition, they are compared in theory and experiment.

After that, there is a comprehensive overview of the current (from 1 December 2017 to now) ship detection algorithms based on deep learning. This part first analyzes the datasets, country, timeline, deep learning framework, and the performance evolution of the 177 papers. The basic situation of 10 datasets in this field is introduced especially. The 177 papers were classified, and they are two-stage, single-stage, anchor free, train from scratch, oriented bounding box, multi-scale, attention model, real-time detection, and so on. The specific algorithms in those papers are analyzed, including the principle, innovation, performance, and the summary.

Finally, the problems existing in this field and the future development direction are described. The main ideas are to design the detection algorithm according to the specific characteristics of SAR image, focus on the detection algorithm without an anchor box, pay enough attention to the detection algorithm of training from scratch, and learn from the existing achievements of natural scene text detection and DOTA, improve the performance of small ships continuously, pay attention to realize the real-time detection of ships through model acceleration and AI chip. It is emphasized that the future important work is to bridge the gap between SAR ship detection and computer vision by merging the existing small datasets into a larger one and making relevant standards.

This review can provide a reference for researchers in this field or researchers interested in this field so that they can quickly understand the current situation and future development direction of this field.

**Author Contributions:** Conceptualization, J.L. and C.X.; methodology, H.S., L.G. and T.W.; investigation, J.L.; writing—original draft preparation, J.L.; writing—review and editing, J.L. and C.X.; supervision, C.X.; funding acquisition, C.X. All authors have read and agreed to the published version of the manuscript.

**Funding:** This research was funded by the National Natural Science Foundation of China, No. 61790550, No. 61790554, No. 61971432, No. 62022092.

**Data Availability Statement:** No new data were created or analyzed in this study. Data sharing is not applicable to this article.

**Conflicts of Interest:** The authors declare no conflict of interest.

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
