# Peer review of "Deep Learning for SAR Ship Detection: Past, Present and Future"

_remotesensing, doi:10.3390/rs14112712_

Round 1
Reviewer 1 Report
The first author of the paper is the pioneer of SAR ship detection. He collected the first dataset and shared it with peers in 2017. Since then, lots of researchers focus their attention on this field. It is very suitable for him to review the past, present and future of this direction. The authors reviewed the 177 papers about SAR ship detection carefully. The advantages and disadvantages in speed and accuracy are also analyzed. The work can make people better understand these algorithms and stimulate more researchers to this field. The knowledge of the authors in this field is relatively comprehensive. They have a correct understanding of the development and research status of this field.
But we think that the following problems should be solved before the publication.
- It is noted that the manuscript needs careful editing by someone with expertise in technical English editing paying particular attention to English grammar, spelling, and sentence structure so that the goals and results of the study are clear to the reader.
- The format of reference [257 XU C, SU H, LI J et al. RSDD-SAR: Rotated Ship Detection Dataset in SAR Images[J]. Journal of Radars.] should be corrected. The following format should be complied with: “Author 1, A.B.; Author 2, C.D. Title of the article. Abbreviated Journal Name Year, Volume, page range.”
- From Table5 to Table 11, AP is used in some cases and AP50 is used in other cases. The author should unify the indicator. The 50 in AP50 usually is superscript (AP50). Besides this, it is suggested to use the same speed index in Table 5, and another column should be added to the parameter quantity. The same problems also exist in tables 6, 7 and 8.
- The summary of the paper is comprehensive, but some classification algorithms are not introduced. Please explain the reason.
- The dataset SMCDD in reference [182] is public recently. And the website is: https://mp.weixin.qq.com/s/PwkkaQaPRUC3ffjlnl7RMg. It should be added in Section 4.3.
- Page4 line157-161, the statement “are not the deep learning based detectors” is reasonable? It should be more rigorous. For example, it can be described as “the detectors are not based on the deep learning”
- The author should add references of pytorch and tensorflow in Page 6 line 237 and line 239.
- It is suggested to describe the advantages of oriented bounding box detection and the difficulties of vertical bounding box detection in Section 5.3.
- The authors introduce the past, present and future of deep learning based SAR ship detection. But the first and the last part is very short, and the second part is very long. It is strange for me. Please explain the reason.
Author Response
We have revised it according to the comments of the reviewers

Reviewer 2 Report
This work reviews SAR ship detection with deep learning. It give a rather long and comprehensive list of most works in this field.
The various databases described are essential for the neural nets according to the general lore that machine learning is 80% data driven and only 20% model driven. The databases should therefore be better described, especially later on in connection with semisupervised learning, where the db has two parts: ships and non-ships. The latter false alarms are not described at all. Yet, they are essential for calculating average precisions AP.
A second concern is the limitations of the review to only ships and mostly chinese publications. There are a number of publications on e.g. TerraSar-X data ship detection: Likewise on classification with e.g. ships and icebergs. The AP's quoted for the different models depend strongly on what the non-ship class is.
More specific comments in lines:
21: how does performance vary "several times"?
75: present refer to published papers which the by definist io past.
85: Ref 70, is later written as 170. Which is it?
119: the distribution may only be known by experts in the field
129: likewise these models are only known by experts. In this case at least define the abbreviation.
131: boosting, is that gradient boosting?
161: what is AIR an abbreviation for?
Table 1: AP=average precision?
Figure 3: "Come into" - do you mean "Entering the"
239: easier and easier..., does that mean that the data is freely available as eg Sentinel-1?
243: Describe Caffe?
260: larger and smaller numbers, do you mean bold face ?
As mentioned above all the AP percentages are strongly data dependent and the data varies greatly in resolution, ship sizes, and what the non-ship class is.
In Fig. 6 NMS is the last step in CNN, but in Fig. 7 NMS is just one of 5 two-stage detectors?
All in all a useful review but unfortunately unclear how to compare the AP for the various databases and networks.
Author Response
We have revised it according to the comments.

Reviewer 3 Report
This paper introduces the past, present, and future of deep learning-based ship detection algorithms in SAR images.
1- Many abbreviations are included in the paper without mentioning them the first time that appears.
2- Fix the typo in Table 4 in the last column.
3- Extensive English is required for the paper, there are many grammatical errors.
4- More discussion is required for the tables in the paper.
Author Response

(The authors gave the same response as above.)

Round 2
Reviewer 2 Report
The authors have clarified a number of my questions. It is, however, not enough just to explain it to the referee. More importantly to the reader who most likely understands less than an expert reviewer.
In several cases the authors refer to explanations in references. A good review can explain it in the text at least briefly so that the reader does not have to look into the hundreds of references. That is the point of a review.
Author Response
We further revised the paper as required.
